# A living biobank of ovarian cancer ex vivo models reveals profound mitotic heterogeneity

Louisa Nelson[1,11], Anthony Tighe[1,11], Anya Golder [1], Samantha Littler[1], Bjorn Bakker[2], Daniela Moralli [3], Syed Murtuza Baker[4], Ian J. Donaldson [4], Diana C.J. Spierings [2], René Wardenaar[2], Bethanie Neale[5], George J. Burghel[6], Brett Winter-Roach[7], Richard Edmondson[1,8], Andrew R. Clamp[9], Gordon C. Jayson[1,9], Sudha Desai [10], Catherine M. Green[3], Andy Hayes [4], Floris Foijer [2], Robert D. Morgan[1,9] & Stephen S. Taylor [1]*

High-grade serous ovarian carcinoma is characterised by *TP53* mutation and extensive chromosome instability (CIN). Because our understanding of CIN mechanisms is based largely on analysing established cell lines, we developed a workflow for generating ex vivo cultures from patient biopsies to provide models that support interrogation of CIN mechanisms in cells not extensively cultured in vitro. Here, we describe a "living biobank" of ovarian cancer models with extensive replicative capacity, derived from both ascites and solid biopsies. Fifteen models are characterised by p53 profiling, exome sequencing and transcriptomics, and karyotyped using single-cell whole-genome sequencing. Time-lapse microscopy reveals catastrophic and highly heterogeneous mitoses, suggesting that analysis of established cell lines probably underestimates mitotic dysfunction in advanced human cancers. Drug profiling reveals cisplatin sensitivities consistent with patient responses, demonstrating that this workflow has potential to generate personalized avatars with advantages over current pre-clinical models and the potential to guide clinical decision making.

[1] Division of Cancer Sciences, Faculty of Biology, Medicine and Health, University of Manchester, Manchester Cancer Research Centre, Wilmslow Road, Manchester M20 4GJ, UK. [2] European Research Institute for the Biology of Ageing (ERIBA), University of Groningen, University Medical Center Groningen, 9713 AV Groningen, The Netherlands. [3] Wellcome Centre Human Genetics, University of Oxford, Roosevelt Drive, Oxford OX3 7BN, UK. [4] Genomic Technologies Core Facility, Faculty of Biology, Medicine and Health, University of Manchester, Michael Smith Building, Dover Street, Manchester M13 9PT, UK. [5] NIHR Manchester Biomedical Research Centre, Manchester University NHS Foundation Trust, Manchester Academic Health Science Centre, Manchester, UK. [6] Genomic Diagnostic Laboratory, St Mary's Hospital, Central Manchester NHS Foundation Trust, Oxford Road, Manchester M13 9WL, UK. [7] Department of Gynaecological Surgery, The Christie NHS Foundation Trust, Wilmslow Rd, Manchester M20 4BX, UK. [8] Department of Gynaecological Surgery, St Mary's Hospital, Central Manchester NHS Foundation Trust, Oxford Road, Manchester M13 9WL, UK. [9] Department of Medical Oncology, The Christie NHS Foundation Trust, Wilmslow Rd, Manchester M20 4BX, UK. [10] Department of Histopathology, The Christie NHS Foundation Trust, Wilmslow Rd, Manchester M20 4BX, UK. [11] These authors contributed equally: Louisa Nelson, Anthony Tighe. *email: stephen.taylor@manchester.ac.uk

Ovarian cancer is the leading cause of gynaecological-related mortality, accounting for ~152,000 deaths worldwide annually[1]. The most prevalent subtype, high-grade serous ovarian carcinoma (HGSOC), which is believed to originate from the fallopian tube[2–5], is particularly lethal because it develops rapidly and often presents with advanced stage disease. Treatment options are limited, typically cytoreductive surgery and platinum/paclitaxel-based chemotherapy[6]. While many patients initially respond well, most develop recurrent disease, yielding relatively poor survival rates that have not changed substantially for 20 years[7].

HGSOC is characterised by ubiquitous *TP53* mutation and extensive copy number variation[8,9]. Recurrent amplifications of *MYC*, *PTK2* and *CCNE1* are common, whereas *PTEN* is frequently lost, and chromosome breakage events often inactivate *NF1* and *RB1*[10–12]. *BRCA1/BRCA2* are inactivated in ~20% of cases, leading to homologous recombination (HR) defects[10], but DNA damage repair defects are more widespread[12,13]. Extensive copy number variation implies chromosomal instability (CIN), i.e. the gain/loss of chromosomes and/or acquisition of structural rearrangements[14]. While p53 loss permits CIN, the underlying primary causes remain poorly understood and are likely complex[15–17]. Indeed, whole-genome sequencing of HGSOCs identified multiple CIN signatures, including foldback inversions, HR deficiency and whole-genome duplication[18,19].

CIN presents both challenges and opportunities when treating HGSOC. By driving phenotypic adaptation, CIN accelerates drug resistance; *ABCB1* rearrangements have been identified in 18.5% of recurrent tumours, enhancing drug-pump-mediated efflux of chemotherapy agents[12,20]. However, CIN can be exploited to develop synthetic-lethality-based strategies, pioneered by the use of poly (ADP-ribose) polymerase (PARP) inhibitors to target *BRCA*-mutant tumours[21–27]. Because of the paucity of actionable driver mutations in HGSOC, synthetic lethality is an attractive option and a better understanding of CIN may open up new therapeutic strategies.

Delineating disease-specific CIN mechanisms and developing novel therapeutic strategies requires models that reflect various human cancers. While judiciously selected cell lines provide tractable models to study cancer cell biology[28], they under-represent the genetic heterogeneity exhibited by tumours[29] and lack the clinical annotations necessary to correlate in vitro drug sensitivities with in vivo chemotherapy responses. While patient-derived xenografts are excellent translational resources[30,31], high-throughput drug profiling is difficult and the timescales involved are challenging in terms of directing personalised treatment. By contrast, living biobanks have the potential to more rapidly generate well-characterised and tractable models suitable for discovery research, drug screening and guiding clinical decisions[32–35]. To develop clinically annotated models that recapitulate HGSOC, we built a living biobank of ex vivo cultures. Here we describe a workflow and exemplar panel of ovarian cancer models (OCMs), and demonstrate their potential to study CIN and drug sensitivity.

## Results

**Establishing a living biobank of ovarian cancer ex vivo models.** To build a living biobank, we established a biopsy pipeline, collecting samples from patients diagnosed with epithelial ovarian cancer treated at the Christie Hospital, and a workflow to generate ex vivo OCMs with extensive proliferative potential. Between May 2016 and June 2019, we collected 312 samples from patients with chemo-naïve and relapsed disease, either as solid biopsies or as ascites (Fig. 1a). Using our standard workflow, thus far we have generated 76 ex vivo cultures. Here, as proof of

principle, we describe 15 OCMs derived from 12 patients. Average patient age at diagnosis was 59 years (range 25–81 years) with a mean survival from diagnosis of 27 months (range 2–125 months; Supplementary Table 1). For 12 samples, ascites were collected following treatment while two ascites and one solid biopsy were chemo-naïve. Ten patients had HGSOC while two had mucinous ovarian carcinoma. Longitudinal biopsies were collected from three patients (Fig. 1a).

To establish cultures, red blood cells were lysed, the remaining cellular fraction harvested by centrifugation, disaggregated if necessary then plated in OCMI media (Fig. 1b). Serial passaging and selective detachment eliminated white blood cells and yielded separate tumour and stromal fractions, which were characterised using phenotypic assays prior to next-generation sequencing and functional profiling. The models are referred to using the OCM prefix followed by the patient number and, if one of a series, the biopsy number (Supplementary Fig. 1a). Models generated independently from the same biopsy are distinguished by an alphabetical suffix. Pilot experiments showed that standard media formulations only supported proliferation of the stromal cells. However, during the course of our pilot studies, Ince et al. described OCMI media which enabled them to establish 25 new patient-derived ovarian cancer cell lines[36]. In our hands, OCMI also supported tumour cell proliferation, allowing us to routinely generate primary cultures with extensive proliferative potential (Supplementary Fig. 1a). Thus our observations confirm the ability of OCMI media to routinely generate ex vivo ovarian cancer models.

**Characterisation of ex vivo models.** To determine whether the OCMs possess the expected hallmarks of ovarian cancer, we characterised the cultures using an array of molecular cell biological approaches (Fig. 1b). Tumour and stromal fractions were morphologically differentiated, with the epithelial appearance of the tumour cells contrasting the fibroblastic stromal cells (Fig. 2a). Time-lapse microscopy and Ki67 expression confirmed both fractions were proliferative (Fig. 2b, c), and the veracity of the separation workflow was confirmed with immunological markers and p53 profiling (Fig. 2d, e and Supplementary Figs. 1a and 2a). Tumour cells were typically positive for PAX8, EpCAM and CA125, and failed to elicit a functional p53 response upon Mdm2 inhibition (Supplementary Fig. 1a). Consistently, tumour cells expressed p53 mutants and frequently overexpressed *MYC* (Supplementary Figs. 1a and 2a). Some tumour cells however were negative for one or more tumour markers despite harbouring *TP53* mutations (Supplementary Fig. 1a), possibly reflecting tumour heterogeneity and/or epithelial–mesenchymal transition[37]. In light of these exceptions, tumour cultures were defined as such if they had an epithelial morphology, expressed PAX8, EpCAM and/or CA125, and/or had a *TP53* mutation, while stromal cells were defined as having a fibroblastic morphology, strong vimentin staining and wild-type *TP53*.

Interestingly, OCM.64–3, generated from the third biopsy from patient 64, exhibited phenotypic heterogeneity; some cells had large, atypical nuclei and were negative for PAX8 and EpCAM, while others were positive for both and had smaller nuclei (Supplementary Fig. 2b). EpCAM/PAX8-positive cells were not detected in OCM.64–1, established from the first biopsy, possibly reflecting tumour evolution during treatment. By exploiting EpCAM status, we separated the two sub-populations (Supplementary Fig. 2c), revealing that only the EpCAM-negative population (OCM.64–3$^{Ep-}$) expressed high levels of MYC (Supplementary Fig. 2a).

Two tumour cultures, OCM.69 and OCM.87, had wild-type *TP53* and a functional p53 response (Supplementary Figs. 1a and

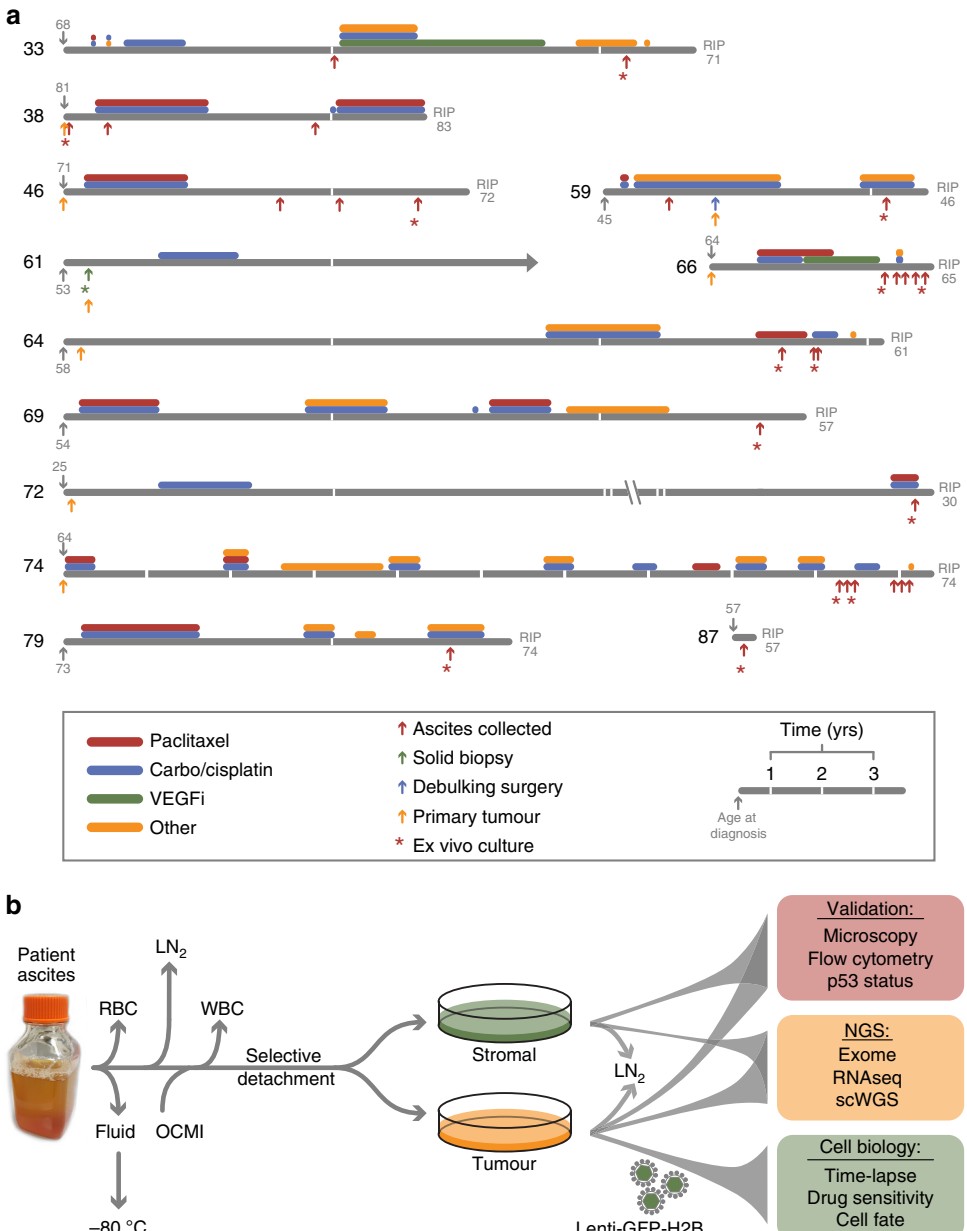

**Fig. 1 Establishing a biobank of ovarian cancer ex vivo models. a** Patient timelines showing age at diagnosis and death, treatments and biopsy collections. **b** Workflow for processing and storage of stromal and tumour fractions. (RBC red blood cells, WBC white blood cells, LN$_2$ and −80 °C specifies long-term storage, OCMI ovarian carcinoma modified Ince medium, NGS next-generation sequencing, VEGFi vascular endothelial growth factor inhibitor). See also Supplementary Fig. 1 and Supplementary Tables 1 and 2.

2a). Re-evaluation of OCM.69, which was also CA125 and EpCAM negative, demonstrated stromal overgrowth so this culture was used as a negative internal control for subsequent studies. By contrast, OCM.87 was positive for PAX8, EpCAM and CA125 and thus confirmed as a tumour model. To determine whether OCMs reflected the primary tumours, we analysed archival tissue, either from the original diagnostic biopsy or from primary cytoreductive surgery (Fig. 1a). Formalin-fixed and paraffin-embedded archival tumour blocks were available for eight patients and immunohistochemistry (IHC) analysis correlated well with immunofluorescence analysis of the ex vivo cultures (Supplementary Fig. 1a, b). For example, OCMs 61 and 72, the two mucinous tumours, were PAX8 negative in both contexts. By contrast, OCMs 46, 66 and the other the HGSOC tumours were PAX8 positive, consistent with a fallopian tube origin. Interestingly, 74, which yielded a PAX8-

negative OCM 9 years later, displayed focal PAX8 staining indicating that heterogeneity already existed in the primary tumour. Nevertheless, these observations demonstrate that the OCM models possess the hallmarks of cancer cells and reflect their respective primary tumours.

**Exome and gene expression analysis.** To determine if the models displayed the genomic features typical of HGSOC, they were interrogated by exome sequencing and RNAseq. Analysis of exome variants showed that sequential cultures from the same patient had similar mutational burdens (Fig. 3a). p53-proficient OCM.87 displayed a highly elevated mutational load, possibly indicating a tumour driven by a mismatch repair defect. By contrast, the well-differentiated mucinous ovarian carcinoma

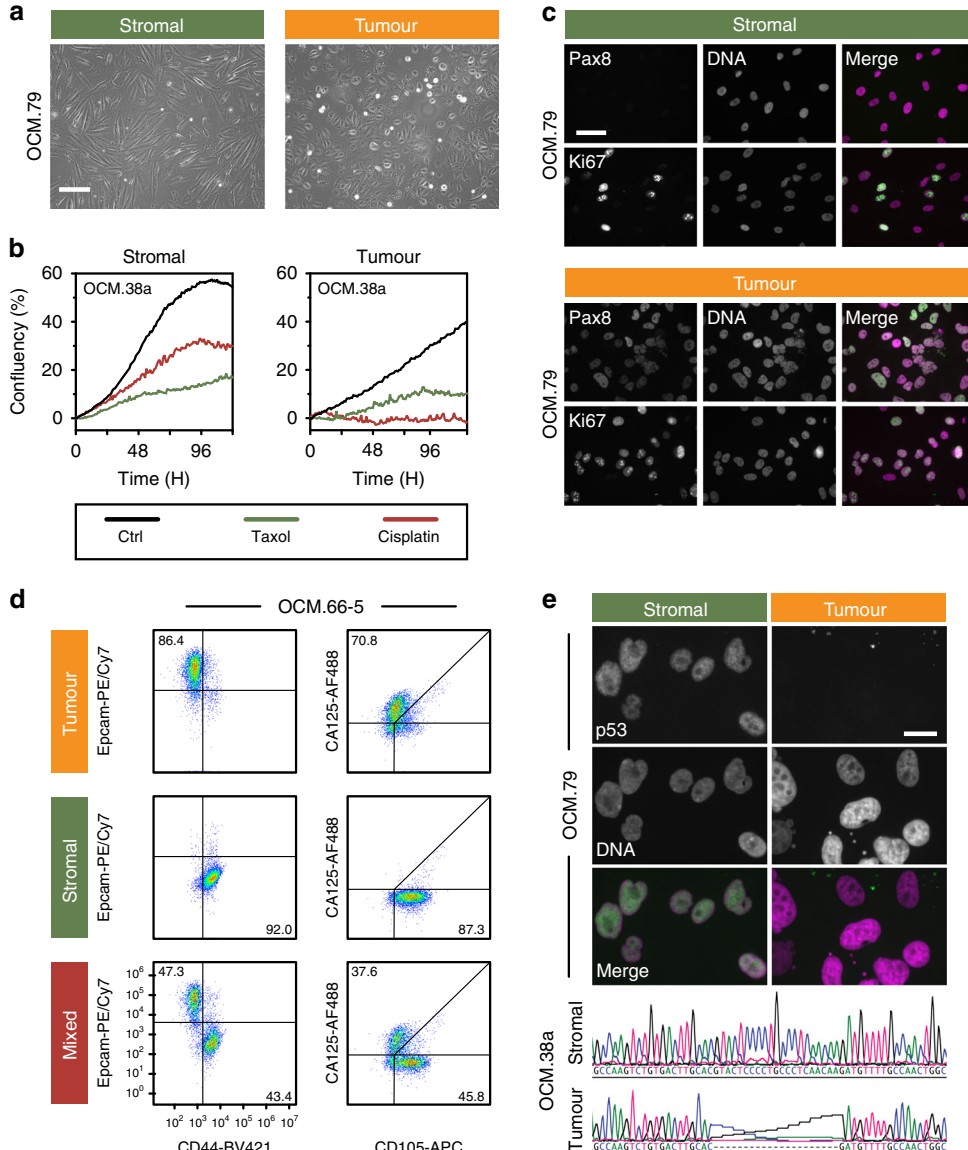

**Fig. 2 Characterisation of ex vivo models. a** Phase contrast images showing distinct morphologies of stromal and tumour cells. Scale bar, 200 μm. **b** Time-lapse imaging measuring confluency showing suppression of proliferation by 1 μM cisplatin and 100 nM paclitaxel. **c** Immunofluorescence images showing expression of PAX8 and Ki67. Scale bar, 50 μm. **d** Flow cytometry profiles quantitating the tumour markers EpCAM and CA125, and the stromal markers CD44 and CD105. Numbers represent percentage of cells in the quadrant. **e** Immunofluorescence images of Nutlin-3-treated cells (OCM.79) showing stabilisation of p53 in stromal cells but not tumour, and DNA sequence showing *TP53* mutation in tumour cells (OCM.38a). Scale bar, 20 μm. Data in panels **a** and **c** are derived from analysis of OCM.79, while data in panels **b** and **d** are derived from analysis of OCMs 38a, and 66-5 respectively. Panels **a**, **c** and **e** are representative images from single experiments. Source data for panels **b**, **c** and **d** are provided as a Source Data file, including the gating/sorting strategy for panel **d**. See also Supplementary Figs. 1 and 2.

model, OCM.61, had a relatively low mutation rate. Interrogating genes known to be mutated in HGSOC confirmed the *TP53* lesions and identified additional mutations in *BRCA1*, *NF1* and *RB1* (Fig. 3b). Importantly, targeted amplicon sequencing of the primary tumours revealed *TP53* mutations identical to those identified by the exome sequencing (Supplementary Table 2), again demonstrating that the OCMs reflect the primary tumours.

Gene expression profiling showed that the tumour and stromal cultures clustered into two distinct clades (Fig. 3c). Principal component analysis (PCA) showed that the stromal cultures clustered very closely, despite originating from 12 different patients (Fig. 3d). While the PCA scores for the tumour cultures associated less tightly, those derived from the same patient, e.g. OCM.66–1 and OCM.66–5, clustered very tightly. The two mucinous cultures

were also closely associated while p53-proficient OCM.87 was an outlier. The phenotypic heterogeneity displayed by OCM.64–3 also manifested in the PCA; OCM.64–3$^{Ep-}$ associated more closely with EpCAM-negative OCM.64–1 but was detached from OCM.64–3$^{Ep+}$. Taken together, these observations further confirm the separation of distinct tumour and stromal populations, and also highlight the phenotypic inter and intratumour heterogeneity.

**Single-cell transcriptomics.** To further explore the phenotypic heterogeneity, we turned to single-cell approaches, initially analysing chemo-naïve OCM.38a using a Fluidigm platform. Hierarchical clustering identified two dominant clusters, Tumour A

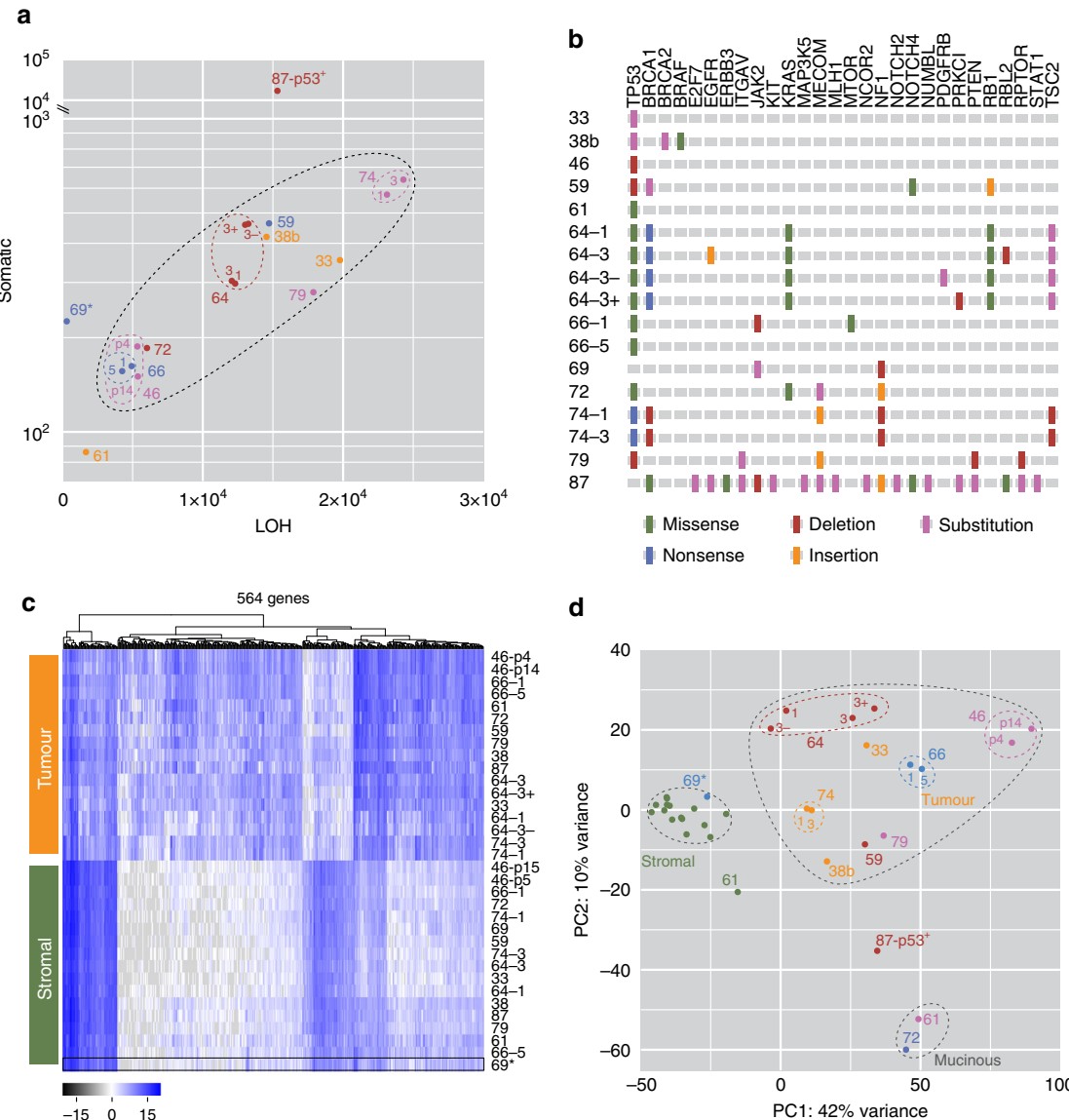

**Fig. 3 Exome and gene expression analysis. a** Whole-exome sequencing showing somatic and loss of heterozygosity variants identified by referencing tumour cells to their matched stromal counterparts. **b** Summary of mutations in genes associated with HGSOC. **c** Hierarchical clustering and **d** principal component analysis of global gene expression profiles, distinguishing stromal and tumour clades, and showing the close relationship of tumour samples from the same patient. 69* is a stromal culture. Symbol colours in **a** and **d** serve to distinguish different OCM tumour samples. Source data for panels **a** and **d** are provided as a Source Data file.

and Stromal A (TA and SA, Fig. 4a). Interspersed within SA were 8 cells from the tumour fraction (TB), presumably contaminating stromal cells. Adjacent to TA was a small cluster from the stromal fraction (SB), possibly reflecting tumour contaminants in the stromal fraction. A PCA and pathway analysis resolved SA into two clusters, SAa and SAb, and SB formed a third, distinct cluster (Fig. 4b). By contrast, TA comprised two overlapping clusters, TAa and TAb. This classification was supported by interrogating specific genes, with the tumour cells expressing *EPCAM*, *TP53* and *MYC* but negative for *XIST* and *TSIX*, consistent with loss of the inactive X chromosome (Fig. 4c). Interrogating cell cycle signatures showed that TAa and TAb had low and high G2/M scores respectively (Fig. 4d). Moreover, genes involved in mitosis and chromosome segregation were overdispersed in the tumour cells (Fig. 4e, f), and the cells expressing high levels of mitotic genes had high G2/M scores (Fig. 4g). Thus, the heterogeneity exhibited by the tumour cells most likely reflects cell cycle stage.

To extend this analysis, we analysed OCMs 38b, 59, 74–1 and 79 using a 10x Genomics platform. Tumour and stromal cells from the four pairs were mixed 3:1 and analysed in parallel. t-SNE plots showed that the majority of cells from each sample formed distinct clusters, whereas smaller fractions formed an overlapping cluster (Fig. 5a). Based on the 3:1 mix, we reasoned that the large distinct clusters represented the tumour cells while the overlapping cluster corresponded to the stromal cells. Consistently, the distinct clusters accounted for ~75% of the cells while ~25% made up the overlapping cluster (Fig. 5b). Moreover, cells in two of the distinct clusters did not express XIST, consistent with loss of the inactive X chromosome. Pathway analysis identified 10 different sub-clusters (Fig. 5d). Seven were private to the tumour cells, with OCMs 38b and 79 dominated by single sub-clusters (1 (87%) and 2 (96%)), OCM.74–1 composed of two sub-clusters (3 (69%) and 7 (30%)), and OCM.59 composed of three (6 (46%), 8 (17%) and

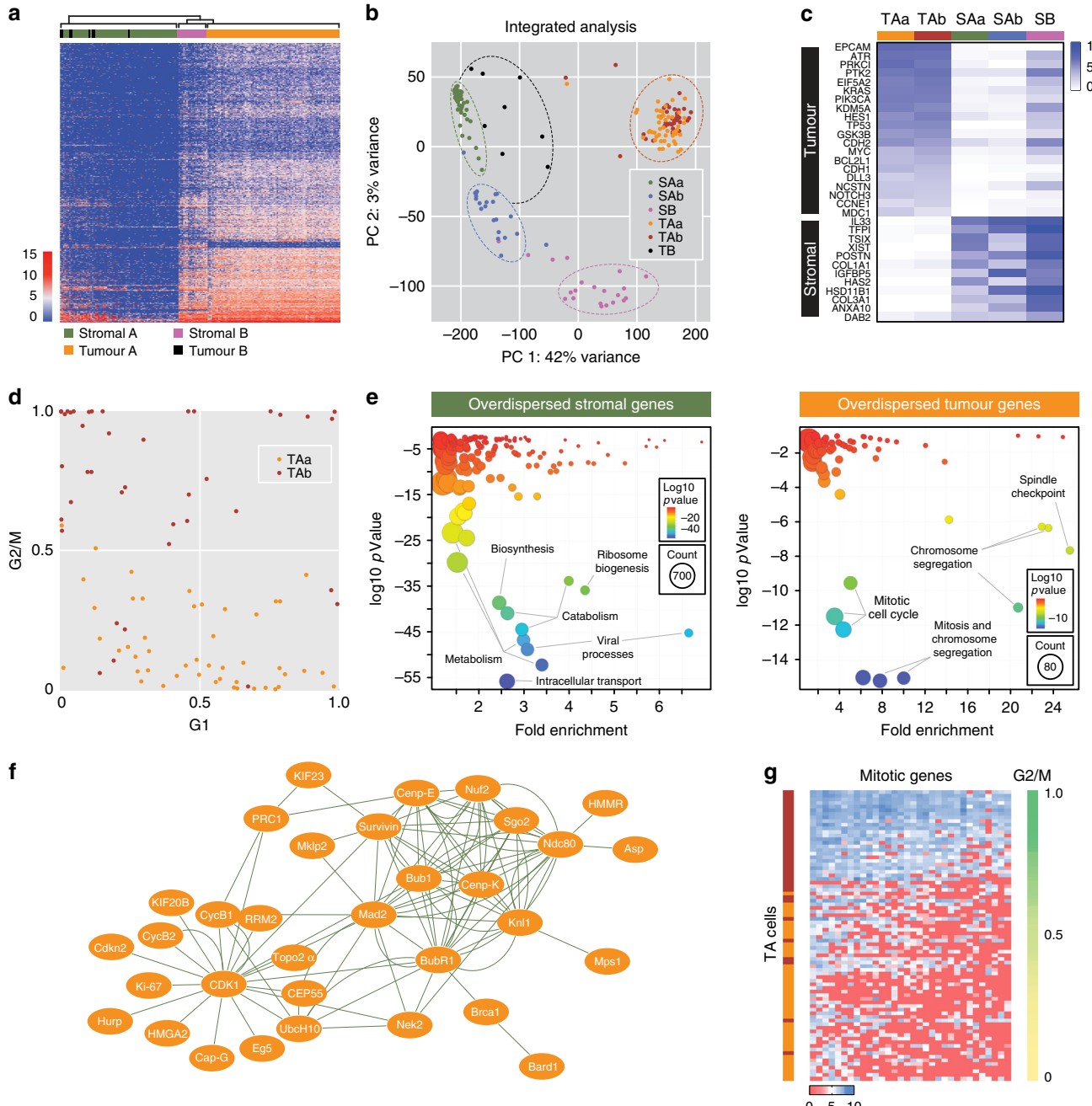

**Fig. 4 Fluidigm single-cell transcriptomics. a** Hierarchical clustering of gene expression profiles distinguishing stromal and tumour cells from chemo-naïve OCM.38a. **b** Principal component analysis integrated with pathway analysis showing subpopulations of tumour and stromal cells. **c** Heat map showing mean expression levels of selected genes in OCM.38a tumour and stromal sub-populations. **d** Scatter plots of G1 score versus G2/M score for individual cells within the TAa/TAb sub-populations. **e** Gene ontology analysis of overdispersed genes in stromal and tumour cells. **f** Network analysis of overdispersed genes in tumour cells. **g** Heat map of overdispersed genes showing that TAb cells expressing higher levels of mitotic genes and have high G2/M scores. Source data for panels **b**–**e** and **g** are provided as a Source Data file.

9 (37%)). By contrast, three sub-clusters were shared between the stromal cells from all four patients; for example, 24%, 42% and 35% of the OCM.38b stromal cells fell into sub-clusters 4, 5 and 10 respectively. Thus, single-cell transcriptomics confirms that despite originating from different patients, the stromal cells are phenotypically similar while the tumour cells display marked inter-patient heterogeneity. Further analysis will however be required to evaluate the nature of this heterogeneity, including whether or not it reflects differences in cell cycle stage. Nevertheless, these data highlight an advantage of deriving

ex vivo models, namely the ability to analyse highly purified tumour fractions unfettered by contaminating stromal cells and the microenvironment.

**Single-cell shallow whole-genome sequencing.** To karyotype the ex vivo models, cultures were subjected to single-cell whole-genome sequencing (scWGS). Analysis of stromal cultures showed that they were largely diploid (Fig. 6a and Supplementary Fig. 3b). By contrast, the tumour cells displayed profound deviations. Moreover, the inter-cellular heterogeneity within any given culture was

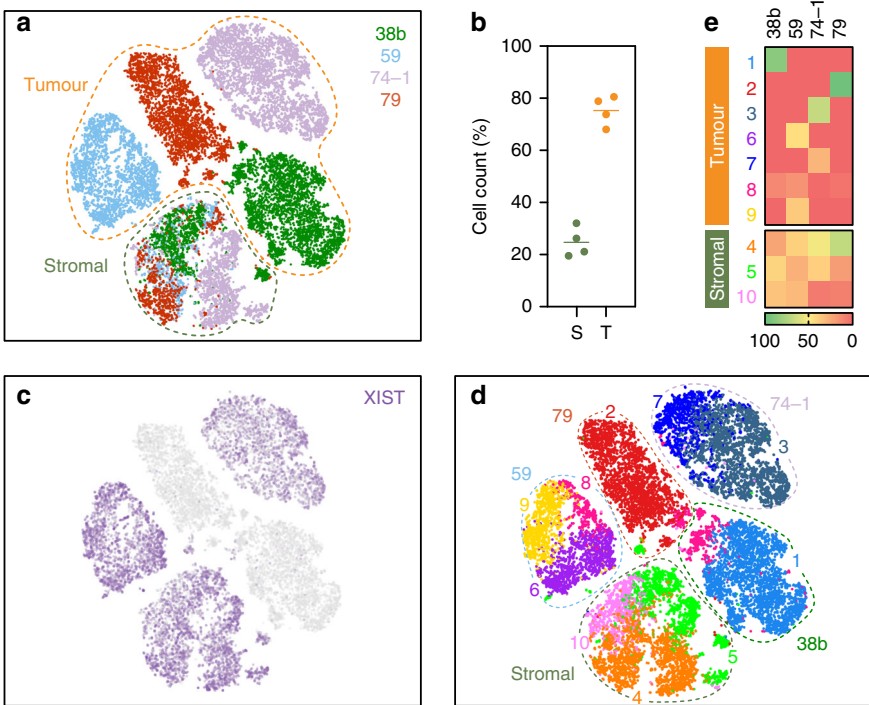

**Fig. 5 10x Genomics single-cell transcriptomics. a** t-Stochastic neighbour embedding (t-SNE) plot showing clustering of single cells from four OCM pairs, with tumour and stromal cells mixed 3:1. **b** Dot plots quantitating the percentage of cells in the stromal and tumour clusters. Line represents the mean ($N = 4$ biological independent samples i.e. $n = 1$ for each of the four OCMs). **c**, **d** t-SNE plot from **a** overlaid with XIST expression (**c**) and 10 sub-populations identified by hierarchical clustering (**d**). **e** Heat maps quantitating the percentage of cells from each patient sample in the 10 sub-populations. Source data for panels **b** and **e** are provided as a Source Data file.

conspicuous, consistent with extensive CIN. Interestingly, four features stood out whereby genomes were marked by whole-chromosome aneuploidies, rearranged chromosomes, monosomies or tetrasomies (Fig. 6a, b and Supplementary Fig. 3b). OCMs 38a, 46 and 79 were characterised by whole-chromosome and chromosome arm aneuploidies (Fig. 6a and Supplementary Fig. 3a). By contrast, OCMs 33, 59 and 66–1 also displayed rearrangements and focal amplifications. OCMs 64–1, 87, 38b and, to some extent, 64–3$^{Ep-}$ displayed numerous tetrasomies, while OCMs 64–3$^{Ep+}$ and 74–1/3 harboured several monosomies (Fig. 6a and Supplementary Fig. 3b). Note that OCM.38a and OCM.38b, independent models developed from the same biopsy sample, had very different karyotypes; whether this reflects intratumour heterogeneity or evolution ex vivo remains to be determined. The two mucinous samples were very different; chemo-naïve OCM.61 was largely disomic but OCM.72 displayed numerous aneuploidies and focal amplifications (Supplementary Fig. 3b). Note that while OCM.61 was derived from a low-grade mucinous adenocarcinoma, OCM.72 was derived from a poorly differentiated tumour, indicating more aggressive disease (Supplementary Table 1). The karyotypes of the OCM.64–3 sub-clones were strikingly different; while 64–3$^{Ep-}$ displayed trisomies and tetrasomies, 64–3$^{Ep+}$ harboured monosomies and disomies (Fig. 6a). Moreover, there was an interesting symmetry; the monosomic and disomic chromosomes in 64–3$^{Ep+}$ were typically disomic and tri/tetrasomic respectively in 64–3$^{Ep-}$. While the relationship between these sub-clones remains to be determined, the scWGS vividly highlights the profound CIN exhibited between and within different ovarian cancer models.

**M-FISH reveals highly rearranged chromosomes.** To verify the CIN highlighted by the scWGS karyotyping, we used two orthogonal approaches, namely multiplex fluorescence in situ hybridization (M-FISH) and quantitation of mitotic spindle poles.

Compared with HCT116, a near-diploid colon cancer cell line, OCMs 38b, 66–1 and 79 were dominated by features consistent with the scWGS, namely tetraploidies, rearranged chromosomes and whole chromosome aneuploidies respectively (Fig. 7a). OCM.59 was also dominated by rearranged chromosomes, including recurrent and unique derivative chromosomes, chromosome fragments, micro-chromosomes, dicentrics and ring chromosomes (Fig. 7b). Interestingly, the primary tumour from patient 59 was notable in that the IHC analysis revealed profound nuclear atypia and multi-nucleated giant cells (Supplementary Fig. 1c), indicating that the extensive CIN observed ex vivo was present in vivo.

Immunofluorescence analysis of the stromal cultures and nine established ovarian cancer cell lines showed that mitotic cells were typically bipolar (Fig. 7c, d). By contrast, multipolar spindles were prevalent in OCM tumour cells. We extended this analysis to include eight additional OCMs generated during the latter part of this study, including three recently described by us[38], thereby including an additional four chemo-naïve models. All eight satisfied the working definition above, i.e. they had epithelial morphologies, were positive for PAX8, and/or had a TP53 mutation. Interestingly, in four out of six chemo-naïve OCMs, multipolar spindles were rare (OCMs 38, 118, 124 and 195), consistent with CIN becoming more pervasive as the disease evolves in response to cytotoxic chemotherapy[12,14]. Nevertheless, the M-FISH and spindle pole quantitation supports the extensive CIN observed by the scWGS.

Quantitating spindle poles also gave us an opportunity to analyse CIN in tumour cells at much earlier passage. Because the selective detachment workflow requires several passages, the ex vivo cultures were typically analysed by passage 10. To analyse earlier stages, frozen unseparated populations were recovered (Fig. 1b) and exposed to the Mdm2 inhibitor Nutlin-3, thereby

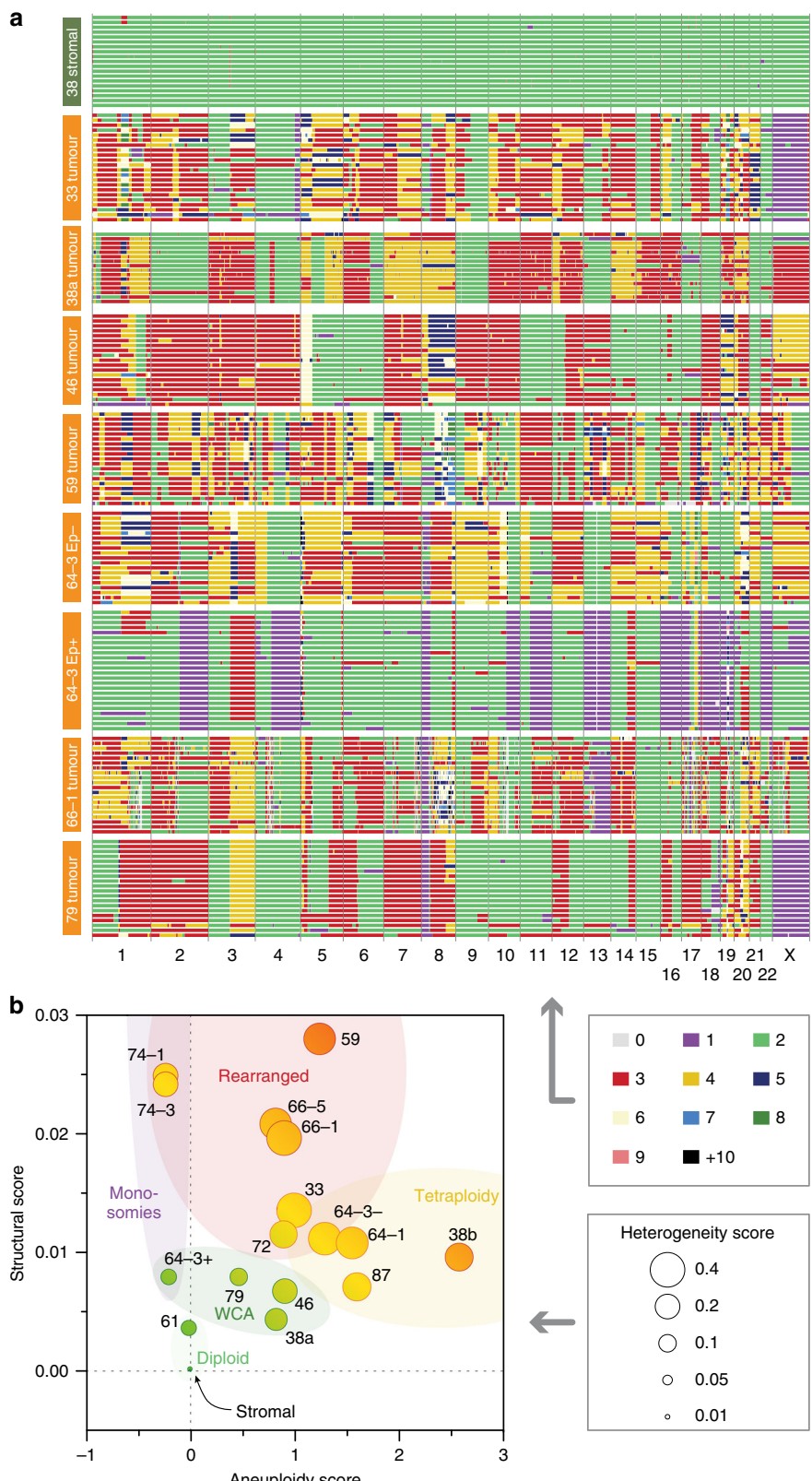

**Fig. 6 scWGS karyotyping. a** Genome-wide chromosome copy number profiles determined by single-cell whole-genome sequencing showing aneuploidies and rearranged chromosomes in tumour cells. Each row represents a single cell, with chromosomes plotted as columns and colours depicting copy number state. **b** Bubble plot showing structural, aneuploidy and heterogeneity scores. See also Supplementary Fig. 3. Source data for panel **b** are provided as a Source Data file.

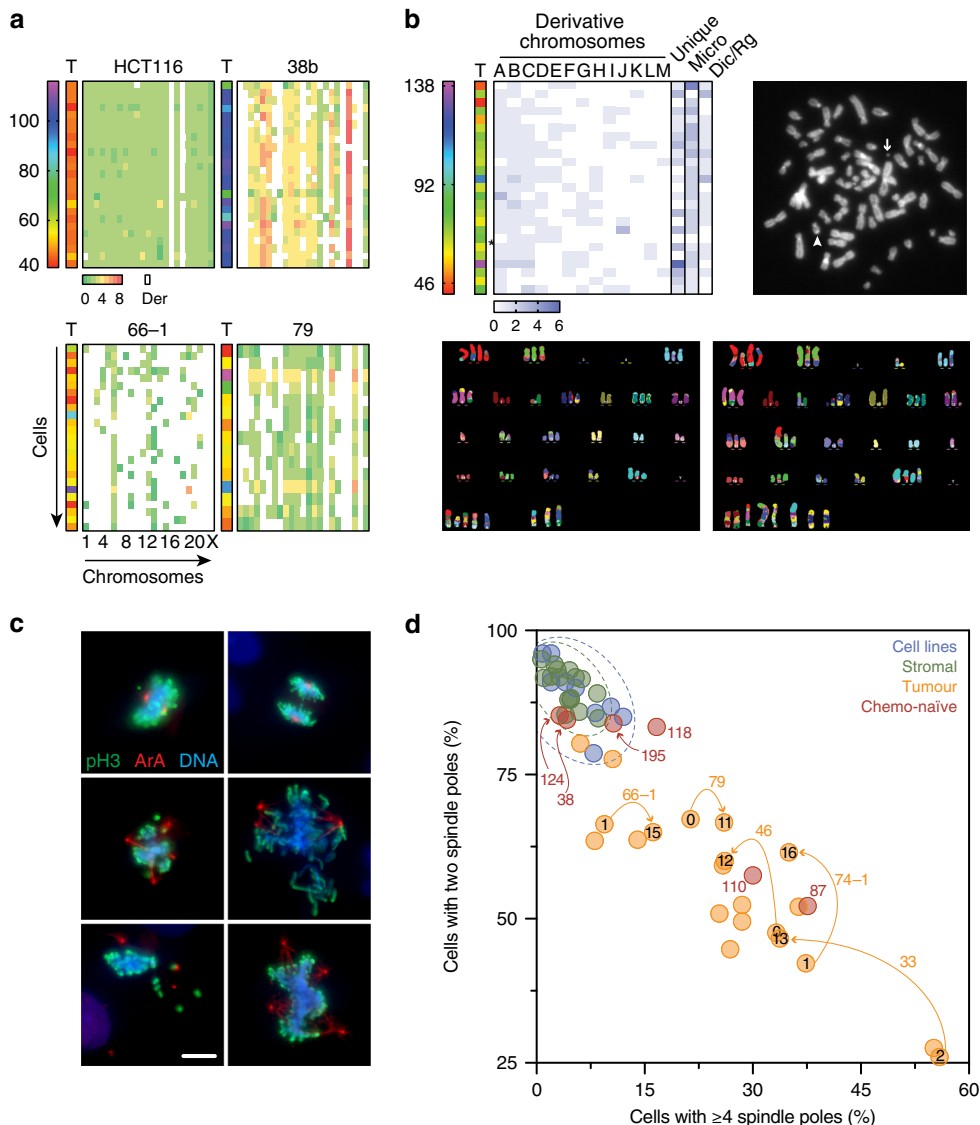

**Fig. 7 M-FISH karyotyping. a** Heat maps quantitating total chromosome count (T, range 40->100) and individual chromosome counts (matrix, range 0–8), for OCMs 38b, 66–1 and 79, enriched for tetraploidy, rearranged chromosomes and whole-chromosome aneuploidy features respectively. Derivative chromosomes indicated by white. HCT116, a near-diploid, stable cell line with two derivative chromosomes, is shown for comparison. **b** M-FISH analysis of OCM.59. Heat map, exemplar chromosome spread and two exemplar M-FISH images. Heat map shows total chromosome count (T) and individual chromosome counts (matrix, range 0–6), quantitating recurring derivatives (A to M), unique derivatives (U), DNA fragments and micro-chromosomes (Micro), and other abnormal structures including dicentrics and ring chromosomes (Dic/Rg). The chromosome spread shows a micro (arrow) and dicentric (arrowhead) chromosome while the M-FISH images show whole chromosome aneuploidies, rearranged chromosomes and different derivatives. **c** Immunofluorescence images of cells stained to detect phospho-histone H3 (serine 10), Aurora A and the DNA (representative images from single experiment). Scale bar, 10 μm. **d** Quantitation of mitotic spindle poles in stromal cells, OCM tumour cells and nine established cell lines. Numbers outside the symbols indicate OCM culture while numbers inside the symbols indicate passage number. Orange arrows connect tumour samples from the same OCM culture analysed at different passages. Source data for panels **a**, **b** and **d** are provided as a Source Data file.

rapidly eliminating the p53-proficient stromal cells. OCMs 33, 46, 66–1, 74–1 and 79 were then analysed between passage zero and two, showing an abundance of multipolar spindles (Fig. 7d). Interestingly, for OCMs 33, 46 and 74–1, the frequency of bipolar spindles increased at later passage, suggesting that continued propagation ex vivo leads to the emergence of relatively stable sub-clones more reminiscent of established cell lines. Nevertheless, our analysis of OCM cells very shortly following biopsy isolation confirms a profound level of CIN, consistent with the scWGS karyotyping.

**Time-lapse microscopy reveals highly abnormal mitoses.** The karyotype heterogeneity and abnormal spindle poles numbers suggests mitotic dysfunction. Indeed, the extensive copy number variations exhibited by HGSOC predicts a high level of CIN. To determine the extent of mitotic dysfunction, we introduced a GFP-tagged histone then characterised the ex vivo models using fluorescence time-lapse microscopy (Fig. 1b). Often, mitosis was successful with chromosomes separating equally (Fig. 8a). Frequently however, chromosome alignment was protracted and segregation abnormal. While stromal cells completed mitosis

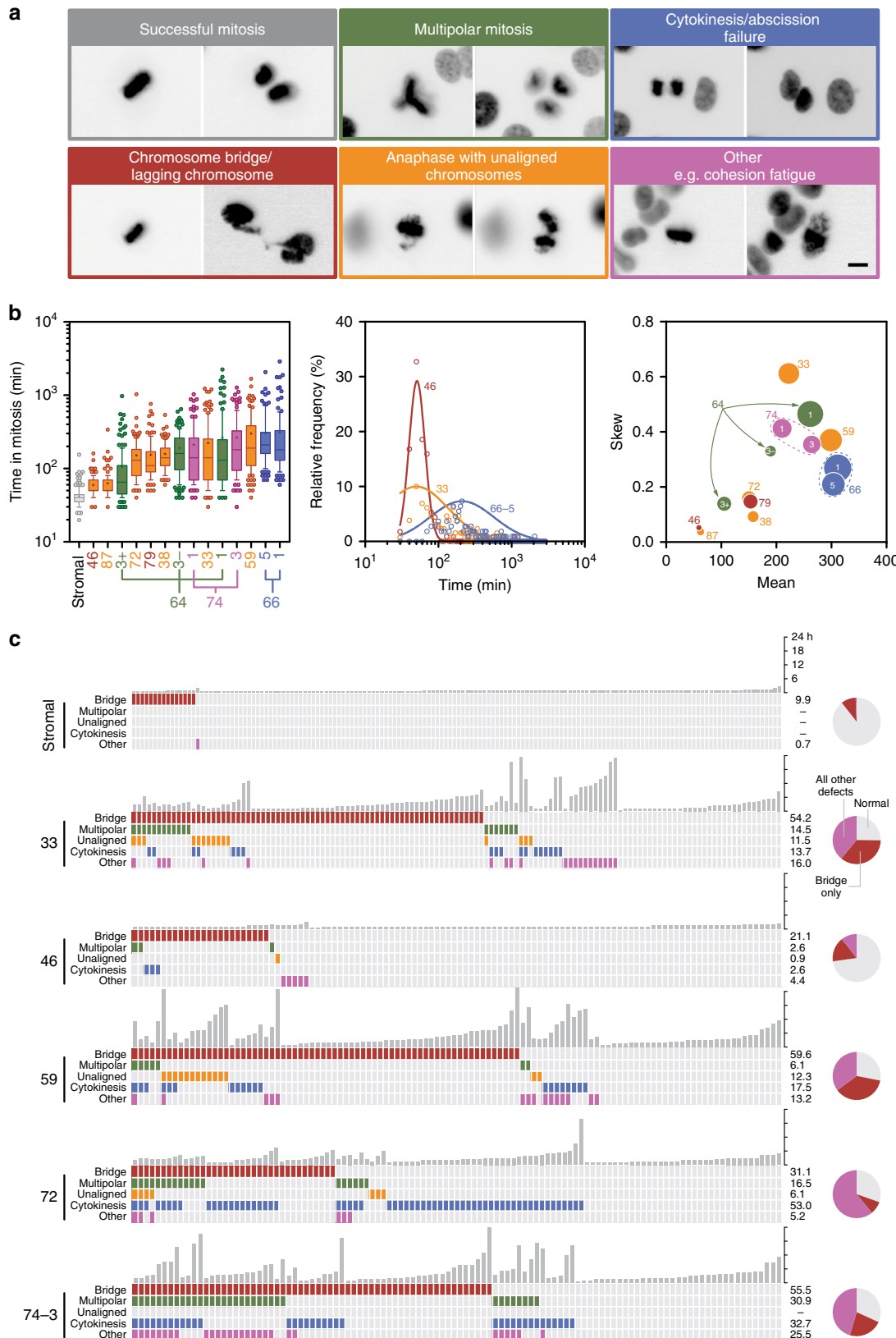

swiftly, mitosis in the tumour cells was protracted and exhibited a profound range, often with skewed distributions (Fig. 8b), consistent with spindle assembly checkpoint (SAC) delaying mitosis[39]. While cultures from the same patient had similar characteristics (e.g. OCM.66–1/5), the OCM.64–3 sub-clones

were dissimilar; OCM.64–3$^{Ep+}$ cells, which have smaller nuclei and monosomies, underwent mitosis faster than their EpCAM negative counterparts (Fig. 8b).

Mitosis in the stromal cells was largely error-free (Fig. 8c). By contrast, lagging chromosomes and anaphase bridges dominated

**Fig. 8 Time-lapse microscopy. a** Examples of abnormal mitoses in tumour cells expressing GFP-H2B, showing images before and after anaphase onset (representative images at multiple positions from single experiment). Scale bar, 10 μm. **b** Analysis of time spent in mitosis, at least 100 cells measured from nuclear envelope breakdown to anaphase onset. Rank ordered box-and-whisker plot with boxes, whiskers and "+" showing the interquartile range, 10–90% range, and mean respectively. Line graph showing linear regression of the frequency distributions for OCMs 33, 46 and 66–5. Bubble plot of Hougaard's skew against the mean, with bubble size proportional to the variance. **c** Quantitation of mitotic anomalies with each column representing one cell and the vertical grey bars representing the time each cell spent in mitosis. Pie charts show the number of normal mitoses, those with anaphase bridges only and all other defects combined. Note that the stromal data is compiled from three cultures, namely OCMs 33, 66 and 79. See also Supplementary Fig. 4. Source data for panels **b** and **c** are provided as a Source Data file, including number of biological independent samples for each OCM in panel B.

the tumour cultures (Fig. 8c and Supplementary Fig. 4a, b), but these events were more dramatic compared with those observed in established CIN cell lines[40,41]. Cytokinesis/abscission failures and multipolar mitoses occurred frequently, with OCMs 72 and 74–3 standing out. Daughter nuclei often reconvened long after anaphase, consistent with DNA blocking abscission. OCM.33 had a high degree of cohesion fatigue[42], possibly accounting for the high skew score (Fig. 8b); note that premature sister chromatid separation prevents SAC satisfaction, enforcing a mitotic arrest[43]. A corollary of this observation is that despite extensive mitotic dysfunction, the SAC is intact. Indeed, cells exhibiting anomalies took longer to complete mitosis and arrested when challenged with paclitaxel (see below). Conversely, OCM.46 completed mitosis relatively quickly and displayed the least number of anomalies (Fig. 8b, c and Supplementary Fig. 4a). However, despite SAC functionality, anaphase with unaligned chromosomes, a phenomenon very rarely seen in established cell lines, was a recurrent feature. OCM.59 stood out with 12% premature anaphases (Fig. 8c). Thus, the time-lapse data demonstrates that mitosis in the ex vivo models is profoundly defective and considerably heterogeneous, indicating that the analysis of established cell lines underestimates the mitotic dysfunction in advanced human cancers.

Disrupting tissue architecture can influence chromosome segregation fidelity[44]. Therefore, we asked whether the OCMs also displayed mitotic dysfunction when cultured as 3D organoids[45]. Analysis of OCM.66–1 in 3D revealed aberrant mitoses including anaphases with unaligned chromosomes (Fig. 9a). We also observed a phenotype not seen in 2D, namely chromosome ejection at anaphase, possibly reflecting the ability of a 3D environment to better anchor ectopic spindle poles. Importantly, the frequency of aberrant mitoses in 3D was similar to 2D (Fig. 9b). Interestingly, the 3D mitoses were not as protracted as those in 2D (Fig. 9c), suggesting that the 3D environment might constrain the spindle leading to more rapid SAC satisfaction.

**Cell fate profiling**. To understand how aberrant mitoses impact cell fate and culture dynamics, we set out to determine proliferation rates and post-mitotic cell fate. Doubling times ranged from under 30 h for OCMs 46 and 87, to over 100 h for OCMs 59 and 74–1/3 (Fig. 10a). Fate profiles of the faster growing models showed that most cells completed multiple cell divisions (Fig. 10b). By contrast, in slow growing OCM.74–1, only 32% of cells divided; 20% remained in interphase and 24% died without entering mitosis. This anti-proliferative phenomenon was observed to some extent in most of the cultures (Supplementary Fig. 5). Taken together with the high frequency of abnormal mitoses described above, a likely explanation is that prior divisions generated daughters harbouring genomes incompatible with continued cell cycle progression. Interestingly, 12% of cells in OCM.74–1 fused with neighbouring cells. Although less frequent, this occurred in several other cultures (Supplementary Fig. 5). Fusion events typically involved daughter cells, suggesting that abscission was not fully executed at the end of the previous cell

cycle[46,47]. Nevertheless, despite the high frequency of abnormal mitoses, sufficient cells survived to yield proliferative cultures.

**Drug sensitivity profiling**. To determine drug sensitivity, we measured culture dynamics in the presence of cisplatin and paclitaxel (Fig. 10c). IC50 values for cisplatin ranged ~7-fold across the cohort, with OCMs 33 and 64–3$^{Ep+}$ the most sensitive and resistant respectively (Fig. 10d). These values did not correlate with paclitaxel IC50 values, which were less variable. While the two cultures from patient 66 responded similarly to both cisplatin and paclitaxel, the two OCM.64-3 sub-cultures diverged considerably, with OCMs 64–3$^{Ep-}$ and 64–3$^{Ep+}$ having cisplatin IC50 values of ~0.6 μM and ~2.1 μM respectively. Despite appearing karyotypically similar, the sequential cultures from patient 74 also had distinct sensitivities, with OCM.74–1 more resistant to both cisplatin and paclitaxel. The patients' tumour responses to chemotherapy broadly correlated with ex vivo drug sensitivities (see Supplementary Table 1). OCMs 33, 38b, and 74–3 had the lowest IC50 values for cisplatin and were derived from patients who achieved a radiological response and a significant reduction in serum CA125 following platinum-based chemotherapy. In contrast, OCMs 46, 59, 64–1, 66–1/5 and 79 originated from patients with progressive disease. Moreover, none of these patients achieved an improvement in serum CA125 levels during treatment. A notable exception was OCM.74–1, which exhibited a cisplatin IC50 suggestive of platinum-resistant disease yet the patient had a partial radiological response and a significant reduction in serum CA125. In this case, the in vivo response could have resulted from the gemcitabine component of her chemotherapy. Nevertheless, the congruence between the patient tumour responses and the drug sensitivity of the ex vivo cultures suggests that models generated by this workflow do indeed reflect the patient's tumours.

**Heterogeneous responses to paclitaxel**. Paclitaxel is routinely used in the treatment of ovarian cancer. Previously, we showed that paclitaxel-induced cytotoxicity in established cancer cells lines is highly heterogeneous[40]. The OCMs also exhibited inter and intra-culture variation (Fig. 10b and Supplementary Fig. 5). For example, in 10 nM paclitaxel, 60% of cells in OCM.46 underwent an abnormal mitosis while at 100 nM, 32% underwent slippage and 22% died in mitosis (Fig. 10b). OCM.87 exhibited a similar behaviour; abnormal mitoses dominated in 10 nM, with 26% slippage and 22% death in mitosis at 100 nM paclitaxel. By contrast, the fate profiles of OCM.66–1 were similar at both concentrations despite an extended mitosis at 100 nM. Consistent with its high IC50, 10 nM paclitaxel had a marginal impact on OCM.74–1, only reducing the number of successful divisions from 32 to 28%. Strikingly in most models, the number of cells that died in interphase following slippage or an abnormal mitosis was low, with an average of only 12% across the cohort. Nevertheless, these observations show that the ex vivo ovarian cancer models represent a valuable resource for drug sensitivity profiling and detailed mode of action studies.

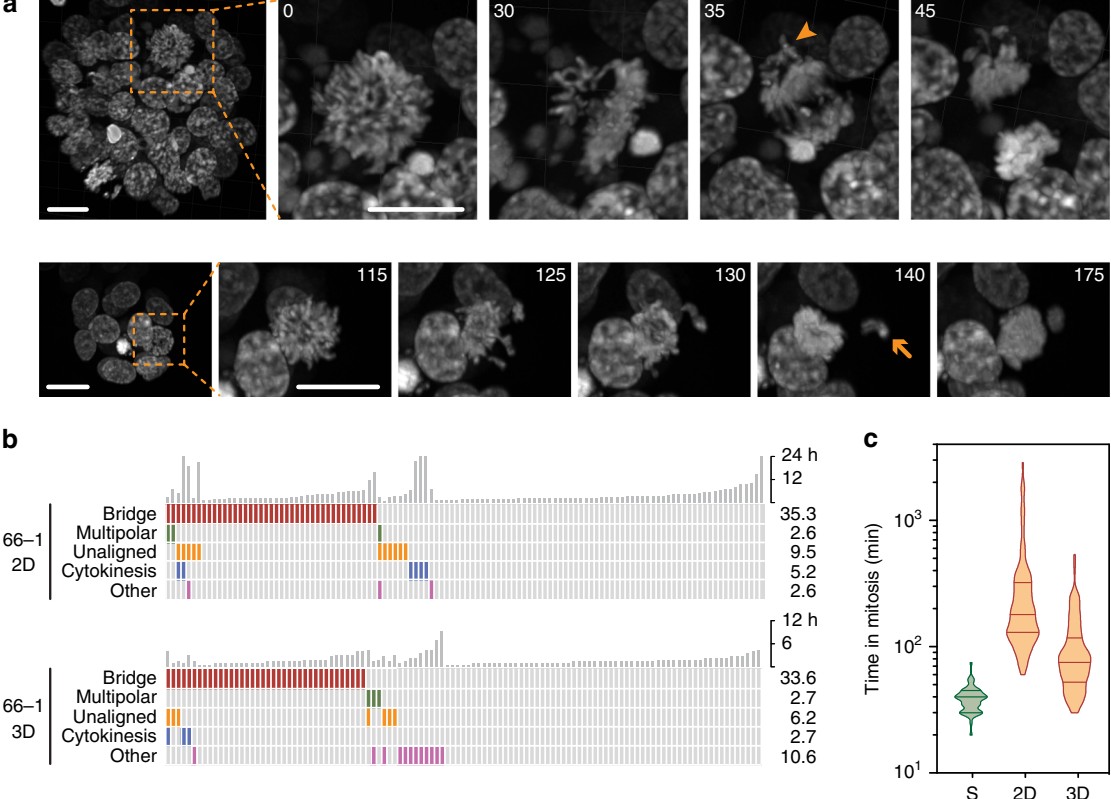

**Fig. 9 Mitosis in 3D. a** Z-stack projections showing examples of abnormal mitoses in OCM.66–1 from three biological replicates when cultured in 3D. Numbers show minutes after imaging initiated. Scale bar, 20 µm. Arrowhead shows unaligned chromosomes at anaphase, arrow shows an ejected chromosome. **b** Quantitation of mitotic anomalies with each column representing one cell and the vertical grey bars representing the time each cell spent in mitosis. **c** Violin plot showing the time spent in mitosis for OCM.66–1 when cultured in 3D. Lines show the median and interquartile ranges. The 2D data from Fig. 8b is for comparison only. Source data for panels **b** and **c** are provided as a Source Data file.

## Discussion

Living biobanks are powerful resources, with the transformative aspect coming from the ability to perform detailed phenotypic studies on well-characterised models that accurately reflect a patient's tumour, and in turn, the ability to correlate ex vivo observations with clinical chemotherapy responses[32–34,48]. As such, living biobanks can potentially address limitations associated with established cancer cell lines, and indeed, our analysis shows that thus far, we have grossly underestimated the mitotic dysfunction in advanced human tumours. The biopsy pipeline and workflow we describe here generates ex vivo ovarian cancer cultures with extensive proliferative potential, rendering models amenable to detailed cell cycle studies, including characterisation of mitotic chromosome segregation and drug sensitivity profiling. Efficient generation of proliferative cultures was facilitated by adopting OCMI media[36], extending the potential of this formulation beyond generating cell lines to also creating tumour cell cultures that can be analysed shortly following biopsy isolation; the vast majority of analyses here were performed within 10 passages. Importantly, by using conditions that allow immediate tumour cell proliferation, bottlenecks that might otherwise select for distinct sub-populations are minimised; indeed, OCMI media maintains the genomic and transcriptomic landscapes of the original tumours[36]. Consistently, the congruence of the gene expression profiles and karyotypes of cultures generated from sequential biopsies indicates that the workflow generates consistent and reflective tumour models. At the same time, the ability of different sub-cultures to emerge indicates that the models also potentially reflect intra-tumour heterogeneity. Important next steps will be to track genomic and phenotypic evolution during culture establishment and propagation. During the course of this work, additional methodologies were described to establish panels of ovarian cancer models, either as 2D cultures and organoids[35,45,49,50]. Another next step will be to compare genome evolution and CIN in these different culture conditions. Moreover, it will be important to characterise the genomes as the primary cultures evolve ex vivo in to established cell lines. The reduction in spindle pole numbers at later passages suggests that more stable subclones might be selected for rapidly once the tumour cells are liberated from the in vivo microenvironment.

The workflow characterising the models involved a complementary array of orthogonal approaches including expression of tumour markers, p53 profiling, exome sequencing, global transcriptomics and scWGS-based karyotyping. Our analysis highlights the risk of relying only on the expression of a small number of tumour markers[51], which is perhaps not surprising in light of the extensive heterogeneity exhibited by HGSOC. And importantly, while the case of OCM.69 highlights the technical challenges during the early phase of culture establishment, it also illustrates the veracity of the workflow. We recognised that this culture was outgrown by stromal cells upon p53 profiling and closer inspection of cell biological parameters. This assessment was confirmed by the exome and RNAseq analysis. Thus far, of the 312 samples from 135 patients, we have attempted to generate cultures from 290, yielding 76 OCMs, i.e., a success rate of 26.2%. These OCMs are derived from 44 patients, yielding a per patient success rate of 32.6%. In some cases, however, when the first attempt failed, we were able to generate a tumour culture from a

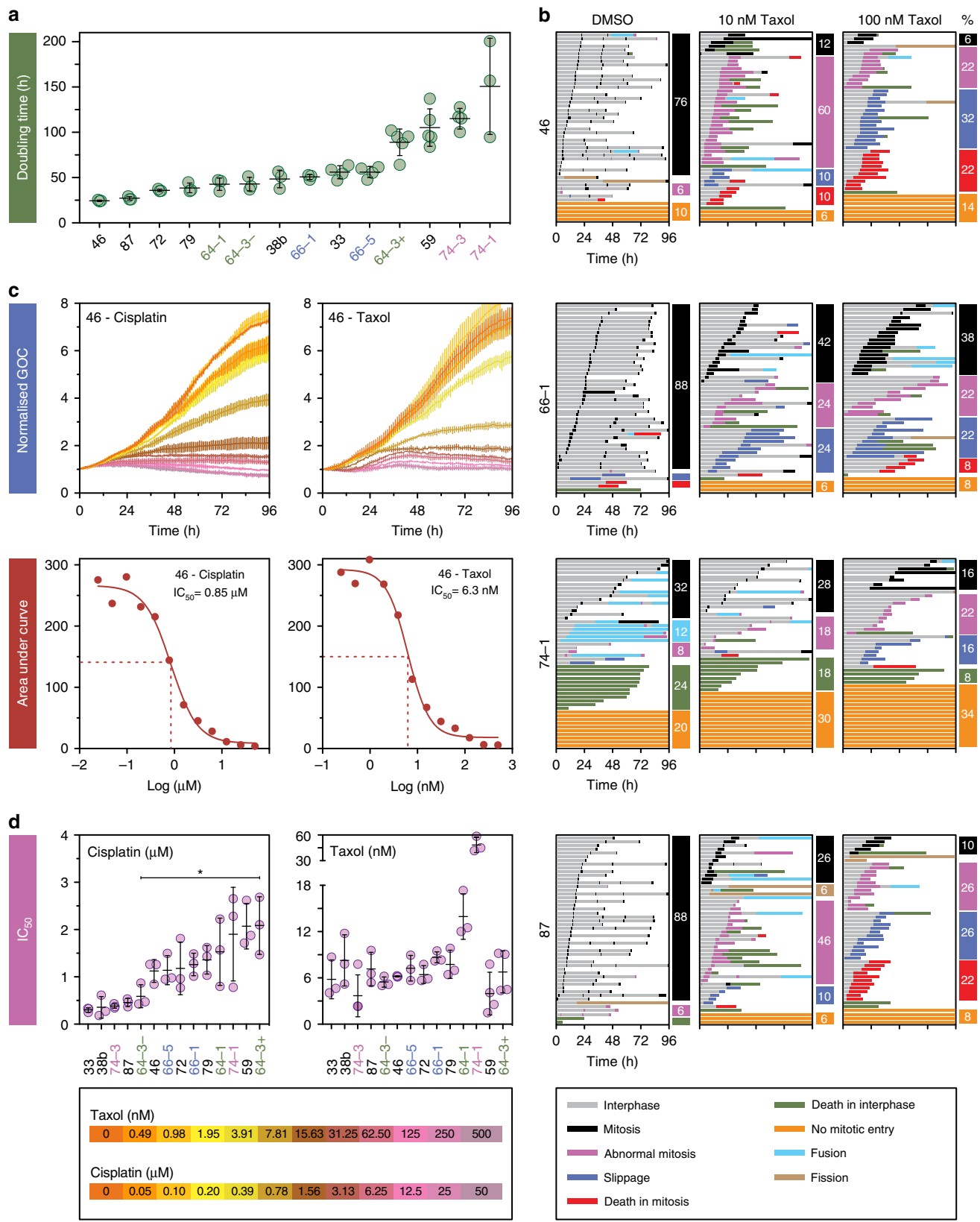

subsequent attempt, facilitated by the availability of frozen, unseparated cells (Fig. 1b). Important next steps will be to define workflow modifications that increase the first-attempt success rate. Preliminary observations suggest that serum source and plating surface can be important factors. All the OCMs described here were generated in low-oxygen conditions, but we note that

several of the cell lines generated by Ince et al.[36] are cultured in atmospheric oxygen, suggesting that oxygen concentration may also be a factor.

The scWGS-based karyotyping was particularly informative, in terms of validating and comparing the different models. In particular, we identified four karyotype features whereby

**Fig. 10 Drug sensitivity profiling. a** Rank ordered plot measuring population doubling times (time-lapse microscopy). **b** Cell fate profiles of untreated cultures and following exposure to paclitaxel, with each horizontal line showing the behaviour of a single cell and the columns quantitating specific cell fates. **c** Line graphs using green object count (GOC) to measure nuclear proliferation of sample OCM.46 in response to increasing concentrations of cisplatin and paclitaxel, plus corresponding $IC_{50}$ curves. **d** Dot plots showing $IC_{50}$ values for cisplatin (rank ordered) and paclitaxel. Asterisk represents $p < 0.05$ for comparison of the sensitivity of OCMs 64-3[Ep−] and 64-3[Ep+] to cisplatin (one-way ANOVA; Tukey's multiple comparison). In **a** and **d**, lines represent mean and standard deviation from at least three biological replicates. In **c** lines show mean and standard deviation from three technical replicates. See also Supplementary Fig. 5. Source data for panels **a** and **d** are provided as a Source Data file.

genomes were enriched for either whole-chromosome aneuploidies, rearranged chromosomes, monosomies or tetrasomies. Integrating these classes with recently described CIN signatures is an important future step[18,19]. By comparing the genomes of single cells, the scWGS-based karyotyping also illustrates the profound heterogeneity within the cultures, indicating pervasive CIN. The proliferative nature of the cultures also facilitated M-FISH karyotyping, which identified structures not detected by sequencing, including acentric fragments and ring chromosomes. However, the key advantage of a living biobank is the ability to perform detailed phenotypic studies on early passage tumour cells, and here we show that ovarian cancer cells display an unprecedented level of mitotic heterogeneity. Analysis of established cell lines has not captured this heterogeneity, presumably because long-term cell culture selects the fitter, more stable subclones. Indeed, clonal evolution analysis of established colorectal cancer cells shows that despite persistent chromosome segregation errors, specific karyotypes are maintained[52], and while multipolar spindles were prevalent in the OCMs, established ovarian cancer cell lines typically undergo bipolar divisions. Another advantage of viable cultures is the ability to analyse highly purified tumour fractions unfettered by contaminating, genetically normal stromal cells and the microenvironment. The workflow does however retain matched tumour-associated fibroblasts and can be adapted to retain tumour-infiltrating lymphocytes[53], in turn allowing reconstruction of tumour-microenvironment interactions.

Consistent with the highly deviant karyotypes, mitosis in the OCMs was often highly aberrant. Note however that most of our analysis was performed on cells grown as monolayers. Importantly, it was recently shown that tissue architecture can influence chromosome segregation fidelity[44]. Specifically, mouse epithelial cells in 3D spheroids exhibited very low missegregation rates; but when disaggregated and analysed in 2D, ~7% of cells displayed a lagging chromosome, a level comparable to that displayed by the patient-derived stromal cells analysed in this study. By contrast, the OCM tumour cells exhibited a much higher rate of abnormal mitoses; 52% of the mitoses we analysed were abnormal. Thus, disrupted tissue architecture is unlikely to account for this very high rate of chromosome missegregation. Indeed, when cultured in 3D, OCM.66−1 exhibited a high frequency of aberrant mitoses.

Despite the high frequency of catastrophic mitoses, sufficient daughter cells survive to yield actively proliferating cultures. However, the doubling times are long compared with established cell lines. Several factors contribute to this including long cell cycle times, cell cycle blocks and apoptosis, indicating that the prior cell division yielded a fatal genome. Nevertheless, the fact that many cells survive following highly abnormal divisions indicates that post-mitotic responses are severely compromised, most likely due in large part to loss of p53 function[14]. However, p53-independent mechanisms may also be defective. For example, as well as driving proliferation and biogenesis, *MYC* drives an apoptosis module that sensitises cells to mitotic abnormalities[54,55]. Interrogating the apoptotic machinery in these models is a future priority, as it may open up opportunities to explore pro-survival inhibitors as therapeutics[56].

The workflow we describe here represents a major step forward in modelling ovarian cancer. In 36 months, we generated 76 ex vivo models from 44 patients, yielding a diverse and comprehensive collection, with the exemplar panel described here providing proof of concept. By addressing the limitations associated with established cell lines, these models better reflect the specific diseases of individual patients, and as such the living biobank will serve as a resource to enable discovery research, in particular enabling a better understanding of CIN, genome evolution and tumour micro-heterogeneity. The tractability of the models in terms of drug sensitivity profiling will also provide tools for drug discovery. Indeed, we recently showed that chemonaïve OCMs derived from patients with platinum-refractory disease are sensitive to a first-in-class compound targeting PARG when combined with a CHK1 inhibitor[38]. A key future priority will be to correlate the drug sensitivity of the ex vivo cultures with in vivo tumour behaviours, in response to both standard of care chemotherapy and emerging agents, a process that will be facilitated by correlating clinical outcomes with each OCM. While the numbers here are small, initial results in terms of platinum responses are encouraging, suggesting that models generated by this workflow could potentially serve as predictive patient avatars. This in turn will provide opportunities to tailor chemotherapy choices based on phenotyping individual tumours as well as stratifying patients for clinical trials testing new agents.

## Methods

**Patient samples.** Research samples were obtained from the Manchester Cancer Research Centre (MCRC) Biobank with informed patient consent obtained prior to sample collection. The MCRC Biobank is licensed by the Human Tissue Authority (license number: 30004) and is ethically approved as a research tissue bank by the South Manchester Research Ethics Committee (Ref: 07/H1003/161+5). The role of the MCRC Biobank is to distribute samples and does not endorse studies performed or the interpretation of results. For more information see www.mcrc.manchester.ac.uk/Biobank/Ethics-and-Licensing.

**Cell culture.** Ovarian cancer and stromal cells were cultured in OCMI media[36] using a 50:50 mix of Nutrient Mixture Ham's F12 (Sigma Aldrich) and Medium 199 (Life Technologies) was supplemented with 5% FBS (Life Science Group) or 5% Hyclone FBS (GE Healthcare), 2 mM glutamine (Sigma Aldrich), 100 U/ml penicillin, 100 U/ml streptomycin (Sigma Aldrich), 10 mM HEPES at pH7.4, 20 µg/ml insulin, 0.01 µg/ml EGF; 0.5 µg/ml hydrocortisone, 10 µg/ml transferrin, 0.2 pg/ml Tridothyronine, 5 µg/ml o-phosphoryl ethanolamine, 8 ng/ml selenious acid, 0.5 ng/ml 17 β-oestradiol, 5 µg/ml all trans retinoic acid, 1.75 µg/ml hypoxanthine, 0.05 µg/ml lipoic acid, 0.05 µg/ml cholesterol, 0.012 µg/ml ascorbic acid, 0.003 µg/ml α-tocopherol phosphate; 0.025 µg/ml calciferol, 3.5 µg/ml choline chloride, 0.33 µg/ml folic acid, 0.35 µg/ml vitamin B12, 0.08 µg/ml thiamine HCL, 4.5 µg/ml i-inositol, 0.075 µg/ml uracil, 0.125 µg/ml ribose, 0.0125 µg/ml para-aminobenzioic acid, 1.25 mg/ml BSA, 0.085 µg/ml xanthine and 25 ng/ml cholera toxin (all from Sigma). Taxol (Sigma Aldrich) and Nutlin-3 (Sigma Aldrich), dissolved in DMSO, and Cisplatin (Sigma Aldrich), dissolved in 0.9% sodium chloride, were stored below −20 ˚C. Nutlin-3 was used at a final concentration of 10 µM. Taxol and Cisplatin were used as described in the figure legends. Established ovarian carcinoma cell lines COV318, COV362 (Sigma), CAOV3 (ATCC) were cultured in DMEM, while OVCAR3 (ATCC), Kuramochi, OVSAHO, OVMANA and OVISE (JCRB Cell Bank) were cultured in RPMI. RMG1 (JCRB Cell Bank) were cultured in Hams-F12 media. HCT116 colon cancer cells were from the ATCC and cultured in DMEM. All cell lines were grown with 10% foetal bovine serum, 100 U/ml penicillin, 100 U/ml streptomycin and 2 mM glutamine, and were maintained at 37 ˚C in a humidified 5% $CO_2$ atmosphere. OV56 (Sigma) were cultured in DMEM/F12 as above but supplemented with 10 mg/ml insulin, 0.5 mg/ml hydrocortisone and 5% foetal bovine serum. All lines were authenticated

by the Molecular Biology Core Facility at the CRUK Manchester Institute using Promega Powerplex 21 System and periodically tested for mycoplasma.

**Establishment of ex vivo models.** Ascites was centrifuged ($500 \times g$ for 10 min at 4 °C) and cell pellets pooled in HBSS (Life Technologies). Red blood cells were removed using a red blood cell lysis buffer (Miltenyi Biotec) as per the manufacturer's instructions. Tumour cells were plated into Primaria flasks containing OCMI. Solid tumour samples were processed using a tumour dissociation kit (Miltenyi Biotec) following manufacturer's instructions and cells plated into collagen-coated 12.5 cm² flasks containing OCMI. All cultures were incubated for 2–4 days at 37 °C in a humidified 5% $CO_2$ and 5% $O_2$ atmosphere. Media was replaced every 3–4 days. Upon cell attachment, stromal cells were separated from the mixed sample using 0.05% trypsin-EDTA, and plated in gelatin-coated flasks in OCMI media containing 5% FBS. Once tumour cells reached 95% confluency, cells were passaged using 0.25% Trypsin-EDTA, centrifuged in DMEM containing 20% FBS and re-plated at a 1:2 ratio. For long-term storage, cells were frozen in Bambanker (Wako pure chemical). Cell separation using EpCAM microbeads (Miltenyi Biotec) was performed according to the manufacturer's instructions. To generate 3D organoids, 20,000 cells were plated in 40 µl Matrigel in 24-well plate. Once solidified Advanced DMEM/F12 supplemented with 1% penicillin-Streptomycin, 1% HEPES, 100 ng/ml Rspondin, 100 ng/ml Noggin, 50 ng/ml EGF, 10 ng/ml FGF-10, 10 ng/ml FGF2, 1x B27, 10 mmol/L Nicotinamide, 1.25 mmol/l N-acetylcysteine, 1 µmol/l Prostaglandin, 10 µmol/l SB202190, 500 nmol/l A83-01 was added and cultured at 37 °C in a humidified 5% $CO_2$ and 5% $O_2$ atmosphere. Media was replaced every 3–4 days.

**Lentiviral transduction.** AAV293T cells (Agilent Technologies) were transfected with pLVX-myc-EmGFP-H2B, psPAX2 and pMD2.G (Addgene) using $CaCl_2$ (Promega) in DMEM supplemented with 10% Hyclone serum (GE Healthcare) and incubated overnight. Virus was harvested 48 later, centrifuged, filtered then added to tumour cells with 10 µg/ml polybrene (Sigma Aldrich) and the cells centrifuged at $300 \times g$, 30 °C for 2.5 h followed by overnight incubation. Puromycin (Sigma Aldrich) (1 µg/ml) was added 48 h after transduction.

**Cell biology.** For immunoblotting, proteins were extracted by boiling cell pellets in sample buffer (0.35 M Tris pH 6.8, 0.1 g/ml sodium dodecyl sulphate, 93 mg/ml dithiothreitol, 30% glycerol, 50 µg/ml bromophenol blue), resolved by SDS-PAGE, then electroblotted onto Immobilon-P membranes (Merck Millipore). Following blocking in 5% dried skimmed milk (Marvel) dissolved in TBST (50 mM Tris pH 7.6, 150 mM NaCl, 0.1% Tween-20), membranes were incubated overnight at 4 °C using the following antibodies: mouse anti-p53 (DO-1) (Santa Cruz Biotechnology cat#sc-126, 1:1000); mouse anti-p21 (F-5) (Santa Cruz Biotechnology cat#sc-6246, 1:100); rabbit anti-c-myc (Y69) (Abcam cat#ab32072, 1:3,500); sheep anti-Tao1[57] (1:1000). Membranes were then washed three times in TBST and incubated for at least 1 h with appropriate horseradish-peroxidase-conjugated secondary antibodies (Rabbit anti-sheep IgG (HL) HRP, cat#618620; Goat anti-mouse IgG (HL) HRP, cat#G21040; Goat anti-rabbit IgG (HL) HRP, cat#G21234; all Invitrogen). After washing in TBST, bound secondary antibodies were detected using either EZ-Chemiluminescence Reagent (Geneflow Ltd) or Luminata Forte Western HRP Substrate (Merck Millipore) and a Biospectrum 500 imaging system (UVP) or ChemiDoc Touch Imaging System (BioRad). For immunofluorescence, cells were plated on collagen- or gelatin-coated 13 mm coverslips and incubated for 48 h. Cells were washed and fixed in 1% formaldehyde, quenched in glycine, then incubated for 30 min at room temperature using the following primary antibodies: rabbit anti-Mucin-16 (Merck Millipore cat#ABC240, 1:50); mouse anti-EpCAM (VU1D9) (Cell Signaling cat#2929, 1:800); rat anti-CD44 (Calbiochem cat#217594,1:200); rabbit anti-Vimentin (EPR3776) (Abcam cat#ab92547, 1:1,000); mouse anti-pan cytokeratin (C-11) (Abcam cat#ab7753, 1:500); mouse anti-Pax8 (Abcam cat#ab53490, 1:100); rabbit anti-Ki67 (Abcam cat#ab15580,1:2,000); and mouse anti-p53 (DO-1) (Santa Cruz Biotechnology cat#sc-126,1:1,000). Coverslips were washed twice in PBS-T (PBS, 0.1% Triton X-100) and incubated with the appropriate fluorescently conjugated secondary antibodies (Donkey anti-Rabbit Cy3, cat#711-165-152; Donkey anti-Rat Cy3, Cat#712-165-153; Donkey anti-Mouse Cy3, Cat#715-165-150; Donkey anti-Mouse Cy2, Cat715-225-150; all Jackson ImmunoResearch Laboratories Inc.) for 30 min at room temperature. Coverslips were washed in PBS-T and DNA stained for 1 min with 1 µg/ml Hoechst 33258 (Sigma Aldrich) at room temperature. Coverslips were further washed in PBS-T and mounted (90% glycerol, 20 mM Tris, pH 9.2) onto slides. Slides were stored at −20 °C prior to image acquisition. Note, for analysis of spindle poles, cells were plated on collagen or gelatin-coated 19 mm coverslips, cultured for 48 h then stained with antibodies to detect phospho-Histone-H3(S10) (Merck Millipore cat#06-570, 1:500) and Aurora A ([58] 1:1000). Images were acquired using an Axioskop2 (Zeiss, Inc.) microscope fitted with a CoolSNAP HQ camera (Photometrics). Image analysis was conducted using Adobe Photoshop CC 2015 (Adobe Systems Inc.). For time-lapse imaging, cells were cultured on collagen-coated 35 mm glass bottom dishes (MatTek Corp) then imaged using an inverted microscope (Axiovert 200; Carl Zeiss, Inc.) equipped with an automated stage (PZ-2000; Applied Scientific Instrumentation) and an environmental control chamber (Solent Scientific), which maintained the cells at 37 °C in a humidified stream of 5%

$CO_2$. Imaging was performed using a ×40 Plan NEOFLUAR objective. Shutters, filter wheels, and point visiting were driven by MetaMorph software (MDS Analytical Technologies) and images captures using an Evolve® Delta camera (Photometrics). For flow cytometry, tumour and stromal cells were incubated in Accutase® (Sigma) to obtain single cell populations then stained with anti-CA125 antibodies (618 F) (Biolegend cat#666902, 1:25) for 30 min at 4 °C, followed by goat anti-mouse Alexa Fluor® 488 (Molecular Probes cat#a11029, 1:100) for 30 min at 4 °C. Cells were washed in PBS then stained with antibodies against EpCAM PE/Cy7 (Biolegend cat#324222, 1:100), CD44 BV421 (Biolegend cat#338810, 1:100) and CD105 APC (Biolegend cat#323208, 1:100), plus Zombie yellow live dead reagent (Biolegend cat#423103, 1:500) for 30 min at 4 °C. Samples were analysed on a Novocyte flow cytometer (ACEA biosciences) and data analysed using Flowjo® software (FlowJo, LLC). To analyse mitosis in 3D organoids, images were acquired using a CSU-X1 spinning disc confocal (Yokogawa) on a Zeiss Axio-Observer Z1 microscope with a ×40/1.3 Plan-Apochromat objective, Evolve EMCCD camera (Photometrics), motorised XYZ stage (ASI) and an environmental control chamber which maintained the cells at 37 °C in a humidified stream of 5% $CO_2$. The 488-nm and 561-nm lasers were controlled using an AOTF through the laserstack (Intelligent Imaging Innovations (3I)) allowing both rapid 'shuttering' of the laser and attenuation of the laser power. Slidebook software (3I) was used to capture images every 5 min over 90 µm at 2 µm Z-intervals. Movies were analysed with Slidebook, and Imaris (bitplane) software.

**Cell proliferation.** To measure proliferation and to perform cell fate profiling, cells expressing GFP-H2B were seeded onto collagen-coated µclear® 96 well plates (Greiner Bio-One). Drugs were added 24 h post-seeding in fresh media. Cells were then imaged using an IncuCyte® ZOOM (Essen BioScience), capturing nine fields of view per well, either every 1–6 h for proliferation and drug sensitivity measurements, or every 10 min for mitotic cell fate profiling. IncuCyte® ZOOM software was used in real-time to measure confluency and green fluorescent object count. The doubling time for each culture was calculated by performing a log2 transformation of the normalised nuclear count; these data were plotted against time and the inverse slope of the log-phase portion of the graph calculated. For drug sensitivity assays, the Area Under the Curve (AUC) at each drug concentration was calculated and plotted against drug concentration to generate dose-response curves from which $IC_{50}$ values were calculated. For cell fate profiling, image sequences were exported in MPEG-4 format and analysed manually. Note that 0 h on the fate profiles represents when imaging started.

***TP53* genotyping by Sanger sequencing.** RNA was extracted using RNeasy Plus Mini kit (Qiagen) and *TP53* cDNA generated by RT-PCR using Superscript III One Step RT-PCR Platinum Taq HiFi (Thermofisher). PCR products were cloned into a pBluescript SK- vector and transformed into XL1-Blue competent cells. Plasmid DNA was extracted using QIAprep Spin Miniprep Kit (Qiagen) and sequenced using the following primers (5′-CAC CAG CAG CTC CTA CAC CG-3′, 5′-ATG AGC GCT GCT CAG ATA GCG-3′, 5′-CGG CTC ATA GGG CAC CAC C-3′, 5′-TCT TCT TTG GCT GGG GAG AGG-3′). Tumour and stromal sequences were aligned using Seqman Pro (DNASTAR).

**Analysis of primary tumours.** Formalin-fixed and paraffin-embedded (FFPE) archival tumour blocks were analysed by immunohistochemistry by collecting 4 µm sections on Superfrost charged slides. After drying overnight at 37 °C, samples were processed using a Ventana Benchmark immunohistochemistry platform (Roche) with antibodies against p53 (Dako cat#M700101-2, 1:50), Cytokeratin7 (CK7, Dako cat#M701801-2, 1:250), PAX8 (Roche cat#06523927001, 1:100) and WT1 (Abcam cat#ab89901, 1:100). Heat induced epitope retrieval was performed using CC1 (Roche), incubating samples at 95 °C for 36, 52, 40 and 64 min for p53, CK7, PAX8 and WT1 respectively. Antibodies were incubated at 37 °C for 32, 40, 32 and 40 min for p53, CK7, PAX8, and WT1, respectively. p53 and CK7 were detected using Ultraview universal DAB kit (Roche), while PAX8 and WT1 were detected using Optiview universal DAB kit (Roche), all as per manufacturer's instructions. Sections were counterstained using Haematoxylin II (Roche) and bluing reagent (Roche) for eight min, and slides imaged using a Leica DM2500 microscope (Leica Microsystems), using a ×20 objective lens under brightfield and processed using Adobe Photoshop. For *TP53* genotyping, FFPE blocks were assessed for total cellularity and the neoplastic cell content of the sample expressed as a percentage of all nucleated cells on a Haematoxylin and Eosin (H&E) stained slide. A neoplastic cell count of ≥10% was required before undertaking DNA extraction. DNA extraction was performed using the cobas® DNA Sample Preparation Kit (Roche). Tumour from $5 \times 5 \mu M$ unstained pathology slides were available for DNA extraction. Extracted DNA was quantified using Qubit 2.0 Fluorometer (ThermoScientific). Targeted enrichment was performed using the GeneRead Clinically Relevant Tumour Targeted Panel V2 (Qiagen). For somatic variants in *TP53* the target read depth across all coding regions (exon 2 to 9) was a minimum of 350×. Mutations were named according to Human Genome Variation Society guidelines (http://www.hgvs.org/) using reference sequence NM_000546.5. All variant calls were independently reviewed using the BAM files and a genome browser (Integrated Genomic Viewer). At a variant allele frequency ≥4% the call sensitivity was >90% and specificity >95% after manual review.

**RNASeq**. RNA was extracted using RNeasy Plus Mini kit (Qiagen), quantified using a Qubit fluorometer (Life Technologies) and quality/integrity assessed using a 2200 TapeStation (Agilent Technologies). Sequencing libraries were then generated using the TruSeq® Stranded mRNA assay (Illumina, Inc.) according to the manufacturer's protocol. Adaptor indices were used to multiplex libraries, which were pooled prior to cluster generation using a cBot instrument (Illumina, Inc.). The loaded flow-cell was then paired-end sequenced (76 + 76 cycles, plus indices) on an Illumina HiSeq4000 instrument. The output data was demultiplexed (allowing one mismatch) and BCL-to-Fastq conversion performed using Illumina's bcl2fastq software. Unmapped paired-reads of 76 bp were interrogated using a quality control pipeline comprising of FastQC v0.11.3 and FastQ Screen v0.9.2 (Babraham Institute). The reads were trimmed to remove any adaptor or poor quality sequence using Trimmomatic v0.36;[59] reads were truncated at a sliding 4 bp window, starting 5′, with a mean quality <Q20, and removed if the final length was <35 bp. The filtered reads were mapped to the human reference sequence analysis set (hg38/Dec. 2013/GRCh38) from the UCSC browser, using STAR v2.4.2a[60]. The genome index was created using the comprehensive Gencode v23 gene annotation[61]. The flag '–outSAMtype BAM Unsorted' was used for the next step. Samtools v1.4[62] was used to identify properly paired reads and create the BAM format required to count read-pairs into genes using htseq-count v0.6.1p1[63] using the flag '–order = name'. The gene counts for each sample were combined using the Linux bash 'paste' function. A header was added to the resulting file (Gene_ID, plus sample names) and the htseq-count summary footer lines were removed. Normalisation and differential expression analysis was performed using DESeq2 v1.10.0 on R v3.2.3[64].

**Exome sequencing**. Genomic DNA was extracted using Purelink Genomic DNA Mini kit (Life Technologies), quantified using a Qubit fluorometer (Life Technologies) and normalised to a final concentration of 5 ng/μl. Indexed, paired-end, sequencing libraries were then prepared using the Nextera Rapid Capture Expanded Exome enrichment kit (Illumina, Inc.), designed to deliver 62 Mb of expertly selected, expanded exonic content. The exome-enriched libraries were then loaded on to a flow-cell and clusters generated using a cBot instrument. Pooled groups of 12 multiplexed libraries were clustered over three lanes (aiming to generate a predicted >100x coverage of the exomes). The flow-cell was then paired-end sequenced (76:76 cycles plus indices) on the Illumina HiSeq4000 instrument and the output data demultiplexed and converted to.fastq format using Illumina's bcl2fastq software. Unmapped paired-reads of 101 bp were interrogated using a quality control pipeline comprising of FastQC v0.11.3 and FastQ Screen v0.9.2 (Babraham Institute). The reads were trimmed to remove any adaptor or poor quality sequence using Trimmomatic v0.36;[59] reads were truncated at a sliding 4 bp window, starting 5′, with a mean quality <Q30, and removed if the final length was <50 bp. The filtered reads were mapped to the human reference sequence analysis set (hg38/Dec. 2013/GRCh38) from the UCSC browser[65], using BWA-MEM v0.7.15[66]. The -M flag was used to flag secondary reads (multimapped). BWA enforces a minimum read mapping score of Q30. The mapped reads were further processed using samtools v1.4[62], to identify properly paired reads, fixmates and sorted by coordinates. Read groups were added to each read, and duplicates flagged using Picard Tools v2.1.0 AddOrReplaceReadGroups and MarkDuplicates, respectively. Variant calling using samtools mpileup and somatic mutations were identified using Varscan 2 (v.2.4.3)[67,68]. Using the matched stromal cell samples as the baseline, variants were classified as follows. If the tumour and stromal sequences matched but do not match the reference genome, the variant is classified as a "germline" mutation. If the tumour and stromal sequences do not match and there is a significant difference in allele frequency, and the stromal sequence matches the reference genome, then the variant is classified as a "somatic" mutation. If by contrast the stromal variant is heterozygous and the tumour variant is homozygous, then the latter is classified as a "loss of heterozygosity" (LOH) event. SNP and indels were annotated by mapping to the Catalogue of Somatic Mutations in Cancer (COSMIC)[69]. Mutations not described in COSMIC were identified as 'unknown'. Data interpretation was aided by use of the Integrative Genomics Viewer (IGV)[70,71].

**Single-cell transcriptomics - Fluidigm**. Single cells were isolated using the Fluidigm C1 platform (Fluidigm Corporation). Medium (10–17 μm) IFCs were used in conjunction with protocol number 100–7168 (Vers. I1) to simultaneously generate cDNAs from the single cells. Sequencing libraries were then constructed using the Nextera XT DNA library kit (Illumina, Inc.) according to the manufacturers' protocol (P/N 15031942, Rev. c) and modified according to Fluidigm Protocol (P/N 100–7168, Vers. I1). Nucleic acid sequencing (76:76 cycles plus indices) was then performed on the NextSeq500 (Illumina, Inc.) using the NextSeq500 mid-output reagent kit (Illumina, Inc.) and the output data demultiplexed and converted to .fastq format again using Illumina's bcl2fastq software. A pre-processing step was used (trimmomatic v0.36) to trim the adaptor and low-quality reads using TruSeq adaptor. The trimmed reads were then aligned to human reference genome GRCh38.p5 (gencode v24)[61] using STAR aligner (v.2.4.2a)[60]. Reads aligning to genes were then counted using HTSeq (v0.6.1.p1)[63]. This count matrix was then used to analyse the dataset using statistical computing programming language, R (R Core Team). During the analysis we started with 192 cells, 96 cells from each fraction. For quality control of cells and genes, we used the function clean.counts() from SCDE package[72] and set the min.lib.size to 500 to filter out cells expressing fewer than 500 genes. For gene filtering, the min.reads was set to 10 and min.

detected to 5 to retain genes with a minimum read count of 10 in at least 5 cells. This filtering, yielded 89 stromal cells and 96 tumour cells for downstream analysis. We then applied PCA to reduce the dimensions of the data. To refine the classification in the PCA plot we applied PAGODA, a pathway based clustering method[73]. A heatmap was generated by taking 10,000 most highly variable genes. We used Cyclone[74] to classify cells to their respective cell-cycle stages based on gene expression. Gene ontology analysis of overdispersed genes was performed using DAVID 6.8[75] and visualised using REVIGO[76]. The network analysis was performed using Cytoscape 3.4.0[77] with the GeneMANIA app[78]. Physical interaction and pathway databases were interrogated to generate network edges.

**Single-cell transcriptomics – 10x Genomics**. Gene expression libraries were prepared from singe cells using the Chromium Controller and Single Cell 3′ Reagent Kits v3 (10x Genomics, Inc. Pleasanton, USA) according to the manufacturer's protocol (CG000183 Rev A). Briefly, nanoliter-scale Gel Beads-in-emulsion (GEMs) were generated by combining barcoded Gel Beads, a master mix containing cells, and partitioning oil onto a Chromium chip. Cells were delivered at a limiting dilution, such that the majority (90–99%) of generated GEMs contain no cell, while the remainder largely contain a single cell. The Gel Beads were then dissolved, primers released, and any co-partitioned cells lysed. Primers containing an Illumina TruSeq Read 1 sequencing primer, a 16-nucleotide 10x Barcode, a 12-nucleotide unique molecular identifier (UMI) and a 30-nucleotide poly(dT) sequence were then mixed with the cell lysate and a master mix containing reverse transcription (RT) reagents. Incubation of the GEMs then yielded barcoded cDNA from poly-adenylated mRNA. Following incubation, GEMs were broken and pooled fractions recovered. First-strand cDNA was then purified from the post GEM-RT reaction mixture using silane magnetic beads and amplified via PCR to generate sufficient mass for library construction. Enzymatic fragmentation and size selection were then used to optimise the cDNA amplicon size. Illumina P5 & P7 sequences, a sample index, and TruSeq Read 2 sequence were added via end repair, A-tailing, adaptor ligation, and PCR to yield final Illumina-compatible sequencing libraries. The resulting sequencing libraries comprised standard Illumina paired-end constructs flanked with P5 and P7 sequences. The 16-bp 10x Barcode and 12 bp UMI were encoded in Read 1, while Read 2 was used to sequence the cDNA fragment. Sample index sequences were incorporated as the i7 index read. Paired-end sequencing (28:98) was performed on the Illumina NextSeq500 platform using NextSeq 500/550 High Output v2.5 (150 Cycles) reagents. The.bcl sequence data were processed for QC purposes using bcl2fastq software (v. 2.20.0.422) and the resulting .fastq files assessed using FastQC (v. 0.11.3), FastqScreen (v. 0.9.2) and FastqStrand (v. 0.0.5) prior to pre-processing with the CellRanger pipeline. The sequence files generated from the instrument were then processed using 10x Genomics custom pipeline Cell Ranger v3.0.1, which generated the fastq files, aligned those files to the required genome, identified the valid barcodes as cells, counted UMIs and reported a gene by cell count matrix in sparse matrix format. Sequences were mapped to the prebuild hg38 genome provided with Cell Ranger package. We used three measures to identify and remove the low-quality cells. Namely, the library size; the number of expressed genes; and the proportion of reads mapped to mitochondrial genes in all four samples. Cells exhibiting a library size lower than three Median Absolute Deviations (MAD) were filtered out. Also, cells expressing a gene count lower than three MAD were filtered out. For mitochondrial read proportions, we filtered out the cells that displayed a percentage of reads mapping to mitochondrial genes greater than three MAD. After this filtering two cells still showed an outlier count on the library size. These two "cells" were assumed to be muliplets and were excluded from further analyses. Raw counts of the remaining cells were then normalised using the deconvolution-based method[79] and then log-transformed. We also filtered out the genes with average counts below 0.01 assuming these low-abundance genes to be unreliable for statistical inference[80]. For visualisation and clustering we first selected the Highly Variable Genes (HVGs). For this we first decomposed the variance of expression in each gene to technical and biological components and identified the genes as HVGs where the biological components were significantly greater than zero. These HVG genes were then used to reduce the dimensions of the dataset using PCA. T-SNE plots were then generated by taking 1–14 components of the PCA. In order to cluster cells in to putative subpopulations we used the Dynamic Tree cut method that can combine the strength of hierarchical clustering and partition around medoids[81]. This method gave us 10 clusters across all populations.

**scWGS karyotyping**. Single G1 nuclei were isolated by cell sorting then processed for sequencing using a Bravo Automated Liquid Handling Platform (Agilent Technologies)[82,83]. Samples were sequenced on an Illumina NextSeq 450 at ERIBA (Illumina). Unprocessed sequencing reads were demultiplexed using library-specific barcodes and converted into fastq format using standard Illumina software (bcl2fastq version 1.8.4). Demultiplexed reads were aligned to human reference genome GRCh38 using Bowtie2 (version 2.2.4). Duplicate reads were marked and removed using BamUtil (version 1.0.3.). Aligned sequencing reads were analysed and curated using AneuFinder (version 1.4.0)[82] using 1 Mb bins. The generation of the heterogeneity and aneuploidy scores are defined[82]. The structural score is defined as the number of copy number state transitions (within a single chromosome) per Mb, then normalised to the number of cells analysed. Data presented as circos plots were generated using Circa software (OMGenomics).

**M-FISH**. Cells were treated with 50 ng/ml colcemid (Sigma Aldrich) for 6 h, then harvested. Cell pellets were incubated for 10 min at room temperature in pre-warmed (37 °C) buffered hypotonic solution (Genial Helix) followed by 20 min on ice. Samples were centrifuged, resuspended in freshly prepared fixative of metha-nol:acetic acid (3:1) and incubated for 30 min. Samples were further centrifuged and incubated in cold (4 °C) fixative for 10 min at room temperature, re-centrifuged and resuspended in cold fixative, dropped onto glass slides, air dried and stored at room temperature. Slides were then experimenter-blinded and hybridised with the M-FISH probe kit 24XCyte (Zeiss MetaSystems) following manufacturer's instructions, and analysed using an Olympus BX60 microscope for epifluorescence equipped with a Sensys CCD camera (Photometrics, USA). Images were collected and analysed using the Genus Cytovision software (Leica). A minimum of 25 metaphases were karyotyped for each cell line/condition.

**Quantification and statistical analysis**. Prism 7 (GraphPad) was used to deter-mine doubling times, AUCs, $IC_{50}$ values and other statistical analyses.

**Reporting summary**. Further information on research design is available in the Nature Research Reporting Summary linked to this article.

## Data availability

Exome sequencing, RNAseq, single-cell RNAseq and scWGS karyotyping data have been deposited at the EMBL-EBI with the following accession numbers E-MTAB-7225, E-MTAB-7223, E-MTAB-724, E-MTAB-8559 and PRJEB28664 respectively. The data underlying Figs. 2b, d, 3a, d, 4b–e, g, 5b, e, 6b, 7a, b, d, 8b, c, 9b, c and 10a, d, and Supplementary Figs. 2a, c and 4a, b are provided as a Source Data file. All other data supporting the findings of this study are available within the article, the Supplementary information files, or the corresponding author upon request. A reporting summary for this article is available as a Supplementary Information file.

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

## Acknowledgements

We thank the patients for their commitment to research; the MCRC Biobank for the sample collection; and members of the Taylor lab for advice and comments on the manuscript. We also thank Peter March and Gary Spencer for 3D microscopy and histology expertise respectively. This research was funded by Cancer Research UK (C1422/A19842) with additional support from the Wellcome Trust Institutional Strategic Support Fund, NWO-TOP (91215003), the NIHR Manchester Biomedical Research Centre, and the University of Manchester. A.G. is supported by an Irshad Akhtar Memorial PhD Scholarship.

## Author contributions

Methodology, investigation and validation by L.N., A.T., A.G., S.L., B.B., D.M., D.S. and R.D.M.; resources by S.D., B.W.-R., A.C., G.J., R.E., C.M.G., A.H. and F.F.; formal analysis by S.M.B., I.D., B.N., R.W. and G.J.B.; conceptualisation, funding, supervision and writing by S.S.T.

## Competing interests

The authors declare no competing interests.
