## [Peer Review File · Nature Communications]

Reviewers' comments:

Reviewer #1 (Expertise: Biobank, organoids, Remarks to the Author):

Nelson et al developed a living biobank of ovarian cancer and demonstrated its application to personalized medicine. The authors performed a large amount of experiments, including establishment of 15 patient-derived HGSOc cell models, transcriptome/WGS profiling and drug screening test. The data supported the conclusion and I feel the manuscript is almost ready for publication in Nature Communication.

I have a few minor requests.

1. It is informative if the authors could show the establishment efficiency of OCM from biopsies or patients.

2. Patient #61 showed CR after the Cisplatin treatment. It would be great if the authors include in vitro Cisplatin response of OCM61 in Fig7D.

Reviewer #2 (Expertise: Ovarian cancer, models, Remarks to the Author):

This manuscript describes the establishment and characterization of 15 primary ovarian cancer cell and matched stromal models, predominantly high grade serous with all but one derived from ascites. The tumor cell models are characterized for chromosomal changes, mutations, gene expression, chemosensitivity and mitoses.

The manuscript is well written, easy to follow, methods are comprehensive and overall will be of interest to the ovarian cancer research and tumor microenvironment communities.

A previous paper referenced in the manuscript (Ince et al, 2015) described the establishment and characterization of ovarian cancer cell lines using the same media. The current paper used ascites rather than tissue and different but overlapping aspects were characterized. Another publication recently described the establishment of 18 ovarian cancer cell models from ascites (Thu et al 2017, PMID: 28881577). This paper which also compares the chromosomal copy number of cell models to the primary tumor is not referenced in the current manuscript. The authors of this manuscript suggest that cell lines "... underrepresent the genetic heterogeneity exhibited by tumours ..."

(Introduction p4, line 64). However, they do not compare the cell models with cell lines. This was done in ovarian cancer cells lines and ascites from patients with ovarian cancer by Penner-Goeke et al, 2017 (PMID: 28376088) who also show the changes in chromosomal instability longitudinally. This paper is not referenced in the current manuscript. The current manuscript also describes generation of matched tumor and stromal models. The novelty of the current manuscript is in the detailed characterization of the tumor models and the generation of stromal models. They state they have validated their pipeline. However, they did not compare primary tumors with the cell models generated, did not characterize the stromal models, and did not compare the chromosomal and genetic characteristics of the matched tumor and stromal models.

The use of ascites in this paper should be noted in the Summary.

The implications of using ascites rather than tumor tissue should be addressed. These include selection bias, such as the increased likelihood of cells that have undergone EMT. This is illustrated in the results (Figure S1) but not well addressed. This figure shows that only 4 of 14 tumor models demonstrated strong/moderate immunofluorescent expression for Epcam, whereas 9 of 14 tumor models demonstrated strong/moderate expression for Vimentin. This is a common problem for ovarian cancer primary culture. However, Results / characterization of ex vivo models, line 102 states "Tumour cells were typically positive for PAX8, EpCAM and CA125 ...". This is incorrect and a more balanced description and representation of the results and their implications is suggested. Line 119 states " ... thus validating the biopsy pipeline and separation workflow." Validating the pipeline would have required comparison of the cell models with primary tissue which they have not done; suggest rewording. What are the stromal cells (eg, are they cancer associated fibroblasts, immune cells, or endothelial cells)? These do not appear to have been defined, thus cannot be validated.

Differentiating tumor and stromal cells by FACS is also not easy. Figure 2 shows these differences, but in panel D the tumor and stromal cells are not matched, suggestive of difficulties. Matched stromal and tumor cells should be used. While the RNA-seq data shows separation of stromal and tumor models, defining the cells prior to any downstream analyses is critical (shown in Fig1B). A definitive explanation of how this was done, and the difficulties encountered is required.

Stromal cells also differ from tumor cells in terms of their chromosomal and genetic stability. As this paper is describing the chromosomal / genetic instability of the tumor models, an excellent comparison would be to also show the results for the stromal cells. This would add confidence to the stromal classification. A comparison of chromosome copy number between matched stromal and tumor cultures is shown for 38 (Fig 4A and repeated in Fig S4B), but not for any of the other matched pairs. No mutation data are shown for stromal cultures. Again, a comparison of matched pairs would have been highly advantageous.

A major drawback of primary cell models / cultures is the limited life / number of passages of the models. This is not addressed in the results at all and only mentioned briefly (line 334) in the Discussion. Similarly, the percentage of specimens that were successfully established as models is not mentioned. As this paper is describing the generation of models, these are very important parameters and considerations of interest to the readers, especially when they wish to use the methods to establish their own cell models and should be included.

Figure 7B needs an explanation for the second column in each graph (? % positive for each phenotype).

The Discussion mentions a paper that is not published (Pillay et al). This needs to be removed, unless the paper is now published.

Reviewer #3 (Expertise: scRNA seq, transcriptomics, Remarks to the Author):

Summary: In this manuscript, the authors generated 15 paired tumor and stromal (normal) cell lines from 12 ovarian cancer patients, demonstrating the genetic and cellular heterogeneity of ovarian cancer (OC). The molecular characteristics of tumor cell lines, including the expression of proliferation/epithelial/OC markers and the status of p53, were compared with that of their matched stromal cell lines, showing the genetic and phenotypic heterogeneity of OC. The genetic and transcriptomic heterogeneity of tumor cell lines were confirmed by exome-seq and bulk RNA-seq. Analyzing DNA copy number variations profiled by single-cell DNA sequencing revealed inter- and intra-tumor chromosomal instability, which was further validated by M-FISH. The authors observed the over-dispersed expression of mitotic genes in a single tumor cell line, measured by single-cell RNA-seq. The over-dispersion of mitotic genes and chromosomal instability imply mitotic dysfunction, which was validated by time-lapse microscopic analysis. The OC tumor cell lines allowed in vitro drug sensitivity profiling that was broadly correlated with the patients' drug response in vivo.

The present manuscript makes a valuable contribution by generating OC tumor cell lines as an in vitro model of OC. However, the authors did not show that their tumor cell lines recapitulate genetic, molecular, and cellular features of in vivo tumors. Furthermore, as a drug screening platform, the utility of OC tumor cell lines should be carefully compared to recent OC organoids (e.g. PUBMED: 30213835).

Major points:

1. Comparison to in vivo tumors: The validity of patient-derived cancer models comes from their close relationship with in vivo tumors. The authors should demonstrate that their OC tumor cell lines have similar histological and genetic features with the matched tumor samples.

2. OC organoids: The authors failed to cite recent works on OC organoids (e.g. PUBMED: 30213835). They should carefully compare their OC tumor cell lines with OC organoids with respect to an in vitro drug screening platform.

3. scRNA-seq: In the single-cell transcriptomic analysis, the number of samples for each condition (stromal and tumor) is one, which means that all the conclusions derived from scRNA-seq cannot be generalizable and have to be interpreted with a caution. In addition, the platform (Fluidigm C1) is known to have a strong chip-to-chip batch effect. The main observation that mitotic genes exhibit strong cell-to-cell variability in gene expression should be supported by including at least two biological replicates for each condition.

4. Bulk RNA-seq: Inferring the molecular subtype (e.g. PUBMED: 30084834) and signature of OC would be helpful for understanding the transcriptomic diversity and drug sensitivity of OC tumor cell lines. The bulk RNA-seq data can be integrated with the Cancer Genome Atlas (TCGA) to infer OC molecular subtype and signatures, which might be useful for interpreting substantial diversity of drug sensitivity.

Minor points:

1. Figure 2: It might be useful if the cell type category of each marker (e.g. CD44: stromal cell marker) is explained.

2. Figure 3A: The "LOH" and "Somatic" should be defined.

3. Figure 3B: The column labels are overlapped.

3. Figure 4B: Each score should be defined.

4. Figure 5C: "Fold enrichment" is not defined.

5. Figure 5D: What does the edge mean? How is the network constructed?

Reviewer #4 (Expertise: CIN, cell cycle, cancer, Remarks to the Author):

Review report Nelson et al.

In this manuscript, the authors describe a pipeline for tumor cell isolation, ex vivo culturing and characterization for ovarian cancers. This 'living biobank' was further exploited to test the behavior of ex vivo cultured tumor cells in mitosis, cell fate and to assess their drug sensitivities. This paper consists out of technically very challenging experiments and the data seems very solid and are presented in a pleasant and clear manner. It involves some of the latest technologies, including single cell sequencing for both DNA and transcriptome, techniques that have not been applied extensively on these type of tumors/cell lines. Setting up living biobanks will be an important aspect of future patient treatment plans and therefore the development and investigation of the feasibility and reliability of such databanks is of great importance. Besides testing drug sensitivities to allow for the predication of therapy responses, these databanks could also provide us with novel biological findings on tumor behavior. Especially, data on the exact extents of instability in tumors and exact numbers on tumor cell heterogeneity are sparse.

Firstly, I highly appreciate the amount of work and the solidity of the data itself. However, I feel that some of the conclusions that the authors draw from this system are a bit premature. Most importantly, I believe the authors missed the opportunity to show that their ex vivo cultures are indeed reflecting their respective tumors in terms of karyotypes, genomes and mitotic behavior. Which is an important feature for future exploitability and for the validity of their main conclusions. IF the authors can address this point, I would support publication in Nature Communication. I will present my main major and minor issues below.

1. In the current manuscript there is no evidence that this pipeline reliably reflects the characteristics and behavior of the tumor itself. If (for at least a small subset) of these patients there is material available from the solid tumor for example from solid biopsies or after surgical resection, this would allow the authors to test the parallels between their system and the original tumor. The fact that the two lines derived from the same biopsy (38a,b) are extremely dissimilar could potentially reflect tumor heterogeneity but could also reflect selection under culture conditions. It is unclear why 38a is not tested in many assays. It would be interesting to see if they have common driver mutations, CIN levels and similar drug sensitivities. If behavior is very different, this could again reflect tumor heterogeneity but could also highlight the potential risks underlying the establishment of ex vivo culturing.

2. Related to the first point, there is no evidence provided that the mitotic behavior observed in this ex vivo culture system reliably reflects the mitotic behaviors in vivo. The 2D ex vivo culturing of these cells might bring extra stresses for these cells that promote segregation errors as these cells are taken out of their 3D context (Knouse et al. 2018) and might even explain the differences observed between these ex vivo cultures and the well established cancer cell lines that have adapted to 2D culturing conditions. And thus, their conclusion that established cell lines underestimate the mitotic

errors that occur in tumors is premature and can only be drawn if data is presented that shows that the ex vivo cultures do actually reflect the mitotic behavior of tumors more reliably (which I understand is a challenging request, so at least their conclusions need to be adjusted and discussion on this matter needs to be included).

3. The original paper describing the OCI medium for the culturing of ovarian cancer cells ex vivo (Ince et al. 2014 Nat Comm) already performed some overlapping analysis; they have shown that drug responses in OCI cultured cell lines largely correlate to patient responses, they have performed transcriptomics and LOH analysis and some basic characterization of their cultured cell lines (p53 mutations, protein expression). So, the novelty of the current manuscript largely lies in the chromosomal instability part/mitotic behavior. As there is no evidence provided that the CIN of the ex vivo cultures reflects that of the tumor (see previous point), this makes me question the overall impact of this manuscript if there is no extra evidence provided on this part. Something that could also enhance the impact is if the authors would attempt to correlate mitotic behavior, CIN levels etc to drug response. If correlations can be identified, this is an extra argument for the importance and relevance of such living biobanks.

Minor issues

1. The timing between treatment and biopsy is highly variable between patients. For example, the sample of patient 38 was treatment naive and only received cisplatin after the ascites collection. In contrast, patient 33 and 69 received platinum relatively long before ascites collection and patient 64 and 74 in between cisplatin treatments. This means that the treatment response was sometimes based on a previous treatment and sometimes on a treatment following cell line establishment. Can the authors indicate these differences?
2. Patient 64 is a bit confusing. The first ascites collection (64-1) was during a taxol treatment while the third collection (64-3) was shortly followed by a cisplatin treatment. Yet, for 64.1 a cisplatin response (negative) is noted while for 64.3 a taxol response is reported. Are the colours of the treatments swapped? Or the data in the table?
3. For patient 61, Figure S1 identified a TP53 mutation by sanger while this was not confirmed by exome sequencing. What could explain this discrepancy?
4. The two sequential biopsies from patient 74 show overlapping mutations but also 3 extra mutations in different genes for the first biopsy as compared to the 3rd biopsy that was taken very soon after the initial one. How do the authors explain this?
5. Figure 2A-C, it is unclear to which cell line these images/graphs belong to.
6. Figure 5C, do the 'overdispersed' genes involve both up- and downregulated genes? Why are they not depicted separately, would this not be more informative?
7. Figure 6B, only 1 stromal cell line is assessed. Which one is this and is the behavior of other stromal cells very consistent? Could the authors analyze a few additional stromal cell lines?

8. Figure 6C. Cancer cell lines often display problems in spindle orientation. When cells do not divide perfectly perpendicular to the substratum they grow on, anaphase behavior is very difficult to score as the metaphase will seem very broad. Could these types of anaphases maybe be scored as 'unaligned' anaphases and thus explain the presence of such events despite a functional SAC?

9. Figure 8B-C, indicate in the figure that these figures relate to OCM59

10. Page 8, line 196; could the overdispersion of mitotic genes not also reflect a higher fraction of proliferative cells?

Response to reviewer comments

We thank the four reviewers for taking the time to evaluate our manuscript. We were delighted with the overall positive response; the critiques were detailed and fair, and in addressing the issues raised, the manuscript has been improved. In the point-by-point response below, we address each specific issue in turn, but first we summarise the major changes to the manuscript, namely the new data sets that have been added:

- 1. Analysis of primary tumours.** Three of the reviewers suggested that we compare the OCMs with their respective primary tumours. This we have now done (Figure S1A and B). Note however, it is important to stress that HGSOC is an incredibly heterogeneous disease, driven by rampant genomic instability. Our scWGS and M-FISH data illustrates this very graphically. Our data also shows evidence of functional heterogeneity, e.g. the two sub-clones within OCM.64-3. Therefore, would we expect that the primary tumours and the OCMs we derive to be identical in every respect? Especially when months if not years, including multiple rounds of chemotherapy, have passed between isolation of the two biopsies. While we might expect the *TP53* truncal mutation to remain constant, *BRCA* reversions are common (Lin *et al.*, 2019) and our analysis of 64-3 shows that Pax8 and MYC status are also plastic. Moreover, the new scRNA-seq data (see below) shows that while three OCMs are represented by a single transcription sub-cluster, OCM.59 is comprised of three different sub-clusters. In addition, in the vast majority of cases, the primary tumours are not killing these women; they are dying due to chemo-resistant recurrent disease, and as Figure 1A shows, our biopsy pipeline and workflow efficiently generates OCMs from chemo-resistant recurrent disease. Taking all this together, the merits of comparing the OCMs with the primary tumours was not immediately obvious to us, and that's why we had not done this analysis for the original manuscript. However, we are glad that the reviewers encouraged us to do this because the congruence between the OCMs and the primary tumours is impressive. In brief, we secured FFPE blocks for 8 of the 12 patients. For the remaining four, there was either no material left or, unlike the ascites, it was not consented for research purposes. Sections were cut and processed for IHC, stained for Pax8, p53, WT1 and CK7, then analysed by Dr Sudha Desai, clinical director for pathology at the Christie and joint lead for gynaecological pathology. When one compares the Pax8 and p53 staining by IHC versus the IF/immunoblotting of the OCMs, the congruence is actually quite remarkable. Moreover, 59 exhibited extensive nuclear atypia and multi-nucleated giant cells; note that OCM.59 has the highest structural aberration score in the scWGS karyotyping. By contrast, nuclear atypia was not prevalent in 61, which has a near diploid karyotype. In addition, DNA was extracted and targeted amplicon sequencing performed by a clinically accredited diagnostic service, analysing 12 genes including *TP53*. Again, the congruence between the Sanger sequencing of RT-PCR products and the targeted amplicon sequencing is excellent. The bottom line therefore is that despite the caveats outlined above, the OCMs do indeed reflect the primary tumours. We have added the results of the IHC and *TP53* sequencing to the table in Figure S1A, and included exemplar IHC images as a new panel, Figure S1B. We also show the *TP53* genotyping in Table S2. We have also modified the timelines in Figure 1A, to show when the primary tumours were sampled, so that the reader can appreciate that in some cases, long time periods and extensive chemotherapy treatments separate the biopsies that gave rise to the IHC data and the *ex vivo* culture.
- 2. Mitosis in 3D.** We now include new data, analysing mitosis in a 3D setting (Figure 9). Specifically, we show that when OCM.66 is grown in 3D, the frequency of mitotic errors is very similarly to that seen in the 2D setting. Interestingly, we now see an additional class of errors not seen in 2D, namely chromosomes being ejected from the spindle at the onset of anaphase.
- 3. Comparison of OCMs with established cell lines.** We include new data to directly compare the OCMs with a panel of 9 established ovarian cancer cell lines. We show that

they are fundamentally different (Figure 5D). In brief, we used spindle pole numbers during mitosis as a proxy for CIN. Consistent with the stromal cells being diploid and mitotically stable, they typically have 2 spindle poles. Consistent with the tumour cells being highly aneuploid and mitotically unstable, they have abnormal numbers of spindle poles. By contrast, despite the established cell lines having abnormal genomes, they typically have 2 spindle poles. This is entirely consistent with a process whereby the long periods in cell culture that gives rise to established cell lines selects for relatively fit and fast growing subclones that are mitotically relatively stable. And again, this underpins the significance of our work. By describing a workflow that allows us to study ovarian cancer cells “*fresh out of a patient*”, we open up new opportunities for discovery science and translational research.

4. **Extension of scRNAseq.** We have extended the single cell transcriptomics. The experiment described in the original manuscript was performed a couple of years ago using a Fluidigm platform. In the meantime, the technology has evolved, allowing us to perform additional scRNA-seq experiments using the more powerful 10x Genomics platform. These new results are present in Figure 7.
5. We have made a number of additional modifications, including repeating some minor experiments, additional bioinformatics analysis, clarification of text/figures, and inclusion of extra citations. We have left in the reference to Pillay et al as this has now been published in *Cancer Cell*, reinforcing the merits of this study by illustrating the potential of the living biobank as a drug discovery platform.

Reviewer #1 (Expertise: Biobank, organoids, Remarks to the Author):

This reviewer is extremely complimentary and we were delighted with their comments. As a biobanking expert, the reviewer is closely aligned with the overall aims of the study and therefore this review carries significant weight.

Nelson et al developed a living biobank of ovarian cancer and demonstrated its application to personalized medicine. The authors performed a large amount of experiments, including establishment of 15 patient-derived HGSOc cell models, transcriptome/WGS profiling and drug screening test. The data supported the conclusion and I feel the manuscript is almost ready for publication in Nature Communication.

I have a few minor requests.

1. It is informative if the authors could show the establishment efficiency of OCM from biopsies or patients.

Agreed; we have added a section at the start of the results to address this, describing capacity and efficiency.

2. Patient #61 showed CR after the Cisplatin treatment. It would be great if the authors include in vitro Cisplatin response of OCM61 in Fig7D.

Agreed, we really wanted to do this experiment but unfortunately #61, which was our first solid biopsy, is one of a very few that did not recover from the frozen stocks. We have tried repeatedly to re-derive this OCM but to no avail. Therefore we have been unable to analyse this sample with all the assays. The inclusion of passage numbers in Figure S1 allows us to acknowledge this.

Reviewer #2 (Expertise: Ovarian cancer, models, Remarks to the Author):

We thank Reviewer 2 for their positive comments; their critique is detailed and fair, and in addressing the issues they raised, the manuscript has been improved. In a few instances, perhaps because we did not explain everything with sufficient clarity in the original manuscript, there do appear to be a few misunderstandings. Hopefully the explanations below and clarifications in the text will resolve these issues.

This manuscript describes the establishment and characterization of 15 primary ovarian cancer cell and matched stromal models, predominantly high grade serous with all but one derived from ascites. The tumor cell models are characterized for chromosomal changes, mutations, gene expression, chemosensitivity and mitoses. The manuscript is well written, easy to follow, methods are comprehensive and overall will be of interest to the ovarian cancer research and tumor microenvironment communities.

A previous paper referenced in the manuscript (Ince et al, 2015) described the establishment and characterization of ovarian cancer cell lines using the same media. The current paper used ascites rather than tissue and different but overlapping aspects were characterized. Another publication recently described the establishment of 18 ovarian cancer cell models from ascites (Thu et al 2017, PMID: 28881577). This paper which also compares the chromosomal copy number of cell models to the primary tumor is not referenced in the current manuscript.

We thank the reviewer for bringing this to our attention and have now cited this paper.

The authors of this manuscript suggest that cell lines "... underrepresent the genetic heterogeneity exhibited by tumours ..." (Introduction p4, line 64). However, they do not compare the cell models with cell lines. This was done in ovarian cancer cells lines and ascites from patients with ovarian cancer by Penner-Goeke et al, 2017 (PMID: 28376088) who also show the changes in chromosomal instability longitudinally. This paper is not referenced in the current manuscript.

Again, we thank the reviewer for bringing this to our attention and have now cited this paper.

The current manuscript also describes generation of matched tumor and stromal models. The novelty of the current manuscript is in the detailed characterization of the tumor models and the generation of stromal models. They state they have validated their pipeline. However, they did not compare primary tumors with the cell models generated ...

The reviewer is correct, we did not include an analysis of the primary tumours. As described above, this we have now done (Figure S1A and B).

... did not characterize the stromal models, and did not compare the chromosomal and genetic characteristics of the matched tumor and stromal models.

This however is not correct. We characterized all the stromal cultures by IF, FACS and TP53 genotyping, spindle pole IF, RNAseq and exome sequencing. Note for the exome analysis, the matched stromal cultures provide the baseline for the variant calling in the tumour cells. In other words, the data in Figure 3 shows the mutations in the tumour models relative to their respective stromal cells. We also included scWGS karyotyping data on one stromal culture but have in fact analysed five in total (Figure R1). The single cell RNAseq was performed on a matched stromal-tumour pair and we now include an additional three stromal-tumour pairs (Figure 5). We also characterized mitosis in three stromal cultures using time-lapse microscopy.

The use of ascites in this paper should be noted in the Summary. The implications of using ascites rather than tumor tissue should be addressed. These include selection bias, such as the increased likelihood of cells that have undergone EMT. This is illustrated in the results (Figure S1) but not well addressed. This figure shows that only 4 of 14 tumor models demonstrated strong/moderate immunofluorescent expression for Epcam, whereas 9 of 14 tumor models demonstrated strong/moderate expression for Vimentin. This is a common problem for ovarian cancer primary culture. However, Results / characterization of ex vivo models, line 102 states "Tumour cells were typically positive for PAX8, EpCAM and CA125 ...". This is incorrect and a more balanced description and representation of the results and their implications is suggested.

We have generated OCMs from both solid biopsies and ascites, but yes, in this proof of principle cohort, there is a preponderance of ascites. We do not see this as an issue. On the contrary, comparing the TCGA data and chemo-resistant disease indicates that primary tumours and ascites are largely congruent (Cancer Genome Atlas Research, 2011; Patch et al., 2015). Moreover, draining ascites provides relatively easy access to longitudinal biopsies spanning the patient's treatment journey; and finally, ascites represents a major

source of morbidity/mortality, so studying tumour cells from ascites is clinically very important.

Regarding tumour markers, yes we agree, they can be variable and we explicitly raised this issue in the Discussion (start of 2nd paragraph). Consequently, we used the word “*typically*” very deliberately because most but not all stained positive for EpCAM, CA125 and PAX8, albeit with varying intensities. Moreover, we included 64-3^{Ep-} and 64-3^{Ep+} precisely because it illustrates the phenotypic heterogeneity point very nicely. From the same biopsy, we have two very different subclones, one positive for Pax8 and EpCAM, the other negative. Yet both have the same p53 mutation and provocatively their karyotypes “*mirror*”, indicating a clonal origin. Note that our purpose here is to demonstrate that the “*tumour*” cells are indeed tumour cells with characteristics that one would expect of HGSOc. This we have done and the data set is compelling. Regarding EMT, although it is not germane to the central thrust of the story, we have now mentioned it as a possible explanation for marker variability.

Line 119 states “ ... thus validating the biopsy pipeline and separation workflow.” Validating the pipeline would have required comparison of the cell models with primary tissue which they have not done; suggest rewording.

By validating, what we mean is that the pipeline and workflow generate OCMs that have the features one would expect of ovarian cancer cells. Having said that, as above, we now include data derived from analysing the primary tumour blocks (Figure S1A).

What are the stromal cells (e.g., are they cancer associated fibroblasts, immune cells, or endothelial cells)? These do not appear to have been defined, thus cannot be validated.

This is an interesting question but again, whether it is germane is arguable. The challenge here, and a common problem in the field, is that primary tumour cultures become overgrown with stromal cells. So again, the purpose of our “*validation*” efforts is to demonstrate that the “*tumour*” cells are indeed tumour cells. As alluded to in Figure 1B, the immune cells are removed during the first few days of cell culture when the media is exchanged, simply because they are non-adherent. There is no obvious reason to assume that the remaining stromal cells are endothelial cells but they could in principle be mesothelial cells (Ahmed and Stenvers, 2013; Kipps et al., 2013). The phase contrast images show fibroblastic morphology, something which is backed up by the IF analysis (Figure R2). So they are most likely fibroblasts and consistent with this, they eventually undergo senescence, which the tumour cells do not. Whether they are “*normal*” fibroblasts or CAFs remains to be determined and that’s why we left the text vague; categorically defining what they are would require a more in-depth analysis, which for the purpose of this study we simply could not justify. As above, the point of this section is to demonstrate that what we are calling “*tumour*” cells are indeed tumour cells and what we are calling “*stroma*” cells are indeed not tumour cells. This we have done, and indeed the data are compelling. Knowing that the stromal cells are indeed stromal cells means that they are a useful reference genome for the exome sequencing (see above) and they provide patient-matched “*normal cell*” controls for the mitosis analysis that comes later. And in the future, they will provide a source of material for reconstruction experiments investigating tumour-stromal interactions.

Differentiating tumor and stromal cells by FACS is also not easy. Figure 2 shows these differences, but in panel D the tumor and stromal cells are not matched, suggestive of difficulties. Matched stromal and tumor cells should be used.

In our hands the FACS works quite nicely, hence the thorough analysis presented in Figure S1A. We typically do not analyse the samples prior to separating them into different fractions, simply because it is unnecessary and a waste of cells. What matters is that after the separation process we can demonstrate that we have purified fractions. The purpose of remixing the tumour and stromal was to demonstrate that the FACS parameters are indeed capable of resolving the two cell types, i.e. it was an important technical control. Nevertheless, the reviewer is correct in that because we did not use matched cells in the figure, it might create a false impression for the reader that the experiment is harder than it

actually is. Therefore, we have now included data from an experiment that does use matched tumour and stromal cells; we thank the reviewer for prompting us to do this. But again, I do feel it important to stress that the point of this validation process is to demonstrate that we have resolved the cells into two fractions, tumour and stromal; this we have done - the data set is compelling.

While the RNA-seq data shows separation of stromal and tumor models, defining the cells prior to any downstream analyses is critical (shown in Fig1B). A definitive explanation of how this was done, and the difficulties encountered is required.

Because the downstream analyses are very costly and labour-intensive, we were very careful to define the cells before progressing on to the RNAseq etc. On a day-to-day basis, the different phase contrast morphological characteristics makes it quite easy to distinguish the stromal and tumour cultures – see above and Figure R2. This differentiation was then confirmed by the *TP53* genotyping and the analysis of immunological markers. This was sufficient to give us the confidence to proceed with the RNAseq which, as the reviewer states, shows very nicely the separation of stromal and tumour cells. So the data clearly shows that our workflow is robust. The definitive explanation of how this was done is already described in the manuscript. Regarding difficulties encountered, please note the issues raised in the second paragraph of the Discussion. For example, we explicitly left in our experience with #69 to illustrate that one has to be careful. Expanding on the challenges would be a good service to the field but is more appropriate for a methods chapter. We do of course need to publish the primary work first before we can consider methods chapters.

Stromal cells also differ from tumor cells in terms of their chromosomal and genetic stability. As this paper is describing the chromosomal / genetic instability of the tumor models, an excellent comparison would be to also show the results for the stromal cells. This would add confidence to the stromal classification. A comparison of chromosome copy number between matched stromal and tumor cultures is shown for 38 (Fig 4A and repeated in Fig S4B), but not for any of the other matched pairs. No mutation data are shown for stromal cultures. Again, a comparison of matched pairs would have been highly advantageous.

This issue has been addressed above, but to reiterate; we performed exome sequencing on all the stromal cultures and we have also analysed five stromal cultures by scWGS - their karyotypes are largely normal (Figure R1). We also analysed three separate stromal cultures by time-lapse data, showing that they segregate their chromosomes efficiently. Note that due to space constraints, we combined the data from all three stromal cultures in the Figure 8, something that we make clear in the legend.

A major drawback of primary cell models / cultures is the limited life / number of passages of the models. This is not addressed in the results at all and only mentioned briefly (line 334) in the Discussion.

We 100% agree with the reviewer, a major drawback with primary cell models – up until now at least – has been the limited life span. Indeed, this exposes a massive gap in the field, one that this manuscript goes a long way towards closing. A fundamental tenet of cancer cell biology is that cancer cells are immortal (Hanahan and Weinberg, 2011). So why therefore do primary cell cultures have a limited number of passages? This makes no sense! But the reason is simple: it's because the culture conditions used in the vast majority of cases is inappropriate! We too suffered from this problem; when we started this project, we tried several standard media formulations and while the stromal cells proliferated immediately, the tumour cells clearly did not. We have very nice time-lapse movies showing this. Then the Ince paper was published; we made up and tested OCMI and the results were remarkable; we can now routinely generate *ex vivo* cultures with extensive proliferative potential. The OCMI system is a game changer, and we explain this in the first section of the Discussion. In our hands, when cultured in OCMI, the tumour cells have continued to proliferate for as long as we have passaged them, in many cases beyond 20 passages and those that we have analysed extensively, over 40. By contrast, although the stromal cells also initially proliferate rapidly in OCMI, they undergo senescence by about passage 10,

exactly as one would expect because they are not immortal. We have added the passage numbers to Figure S1A and expanded the text at the end of the first Results section. But the bottom line here is that we have achieved something highly significant. We didn't invent OCMI, that credit of course belongs to Ince *et al*, but by independently confirming their invention, as “*first followers*”, and using it to generate primary cultures rather than cell lines, we open up a powerful methodology to the wider community. As such this manuscript will represent a significant advance. We have tried to convey this exciting aspect of the story better in the revised manuscript.

Similarly, the percentage of specimens that were successfully established as models is not mentioned. As this paper is describing the generation of models, these are very important parameters and considerations of interest to the readers, especially when they wish to use the methods to establish their own cell modes and should be included.

We agree, these are important parameters and we have added these details to the first section of the Results.

Figure 7B needs an explanation for the second column in each graph (? % positive for each phenotype).

Agreed, we have expanded the legend (Now Figure 10).

The Discussion mentions a paper that is not published (Pillay *et al*). This needs to be removed, unless the paper is now published.

This paper has now been published in *Cancer Cell*, reinforcing the merits of this study by nicely illustrating the potential of the living biobank as a drug discovery platform. We initially submitted the two manuscripts in parallel and it is a shame that we could not synchronize them better.

Reviewer #3 (Expertise: scRNA seq, transcriptomics, Remarks to the Author):

Reviewer 3 is also very complimentary, stating clearly that this work “*makes a valuable contribution*”. The critique is fair and we have now addressed all the points they raise, either with additional experiments, changes to the presentation, or in the explanations below. Consequently, the manuscript has been substantially improved and we thank the reviewer for encouraging us to strengthen the work.

Summary: In this manuscript, the authors generated 15 paired tumor and stromal (normal) cell lines from 12 ovarian cancer patients, demonstrating the genetic and cellular heterogeneity of ovarian cancer (OC). The molecular characteristics of tumor cell lines, including the expression of proliferation/epithelial/OC markers and the status of p53, were compared with that of their matched stromal cell lines, showing the genetic and phenotypic heterogeneity of OC. The genetic and transcriptomic heterogeneity of tumor cell lines were confirmed by exome-seq and bulk RNA-seq. Analyzing DNA copy number variations profiled by single-cell DNA sequencing revealed inter- and intra-tumor chromosomal instability, which was further validated by M-FISH. The authors observed the over-dispersed expression of mitotic genes in a single tumor cell line, measured by single-cell RNA-seq. The over-dispersion of mitotic genes and chromosomal instability imply mitotic dysfunction, which was validated by time-lapse microscopic analysis. The OC tumor cell lines allowed in vitro drug sensitivity profiling that was broadly correlated with the patients' drug response in vivo.

The present manuscript makes a valuable contribution by generating OC tumor cell lines as an in vitro model of OC. However, the authors did not show that their tumor cell lines recapitulate genetic, molecular, and cellular features of in vivo tumors. Furthermore, as a drug screening platform, the utility of OC tumor cell lines should be carefully compared to recent OC organoids (e.g. PUBMED: 30213835).

Major points:

1. Comparison to in vivo tumors: The validity of patient-derived cancer models comes from their close relationship with in vivo tumors. The authors should demonstrate that their OC tumor cell

lines have similar histological and genetic features with the matched tumor samples.

As outlined above, we now include data derived from analysis of the primary tumour blocks (Figure S1A and B), confirming that the OCMs have similar histological and genetic features with their respective primary tumour samples.

2. OC organoids: The authors failed to cite recent works on OC organoids (e.g. PUBMED: 30213835). They should carefully compare their OC tumor cell lines with OC organoids with respect to an in vitro drug screening platform.

A fair point; we have now cited the Hill *et al* paper and the more recent paper from the Clevers' lab (Kopper *et al.*, 2019). In addition, we show that these OCMs are amenable to 3D cultures and we analyse mitosis in one of these (see below). Carefully comparing drug sensitivities of the entire OCM panel in 2D and 3D is a noble aspiration and something we hope to look at in the future, but to request this now is – with all due respect – completely unrealistic. This manuscript already represents an enormous amount of work and we present a number of significant advances that warrant publication. Therefore, we suggest that a 2D vs. 3D comparison will be best suited for a future instalment. Meanwhile, we would like to point out that in several manuscripts that analyse 3D models, the drug sensitivity profiling is actually performed in 2D (e.g. (Kopper *et al.*, 2019)). Thus, what we present here is state-of-the-art and critically, the *in vitro* IC₅₀ values correlate with the clinical responses.

3. scRNA-seq: In the single-cell transcriptomic analysis, the number of samples for each condition (stromal and tumor) is one, which means that all the conclusions derived from scRNA-seq cannot be generalizable and have to be interpreted with a caution. In addition, the platform (Fluidigm C1) is known to have a strong chip-to-chip batch effect. The main observation that mitotic genes exhibit strong cell-to-cell variability in gene expression should be supported by including at least two biological replicates for each condition.

The reviewer is correct, we only analysed one stromal-tumour pair. To be fair however, this was clear in the text and in our view, we didn't over generalise our conclusions. Single cell studies are rapidly evolving and the experiment described in the original manuscript was performed a couple of years ago using a Fluidigm platform at a cost of £13k, making biological replicates challenging. Nevertheless, it makes a very valuable contribution in terms of further supporting our separation workflow. Regarding chip-to-chip variation, the interesting observation that mitotic genes are overdispersed in the tumour cells is based on comparing cells analysed in parallel, i.e. on the same chip, so the results are robust and we are confident in our conclusions. In the meantime, the technology has evolved, allowing us to perform additional scRNA-seq experiments using the more powerful 10xGenomics platform. These new results are present in Figure 5.

4. Bulk RNA-seq: Inferring the molecular subtype (e.g. PUBMED: 30084834) and signature of OC would be helpful for understanding the transcriptomic diversity and drug sensitivity of OC tumor cell lines. The bulk RNA-seq data can be integrated with the Cancer Genome Atlas (TCGA) to infer OC molecular subtype and signatures, which might be useful for interpreting substantial diversity of drug sensitivity.

This is an excellent point and something we discussed long before we submitted the manuscript as we were confident that this question would come up. However, it is not as simple as you might expect, for two reasons. The first is relatively trivial; somewhat surprisingly, data from the previous studies (Cancer Genome Atlas Research, 2011; Tothill *et al.*, 2008) is not easily available in a format that has allowed us to perform a direct comparison. The second is more problematic; it is not at all clear to us that the molecular subtypes based on transcriptional profiling are as robust as seems to be generally accepted by the field. Importantly, Way *et al* re-analysed the five largest publicly-available mRNA expression HGSOc datasets and concluded that there are only two robust subtypes, "mesenchymal" and "proliferative" (Way *et al.*, 2016). They suggested that a third subtype may exist but it is more variable across populations and may represent steps along a continuum. Their data is available in a format that has allowed us to extract the gene names and thus we have aligned our data. A clustering analysis (Figure R3) indicates that three of

our OCMs and Kuramochi cells cluster with the proliferative subtype (C2) while seven cluster with the mesenchymal subtype (C1). The others are more ambiguous. However, what is particularly interesting is that 64-3^{Ep+} tightly aligns with C2 but 64-3^{Ep-} more closely aligns with C1. Because 64-3^{Ep+} and 64-3^{Ep-} are derived from the same clone, this suggests that these subtypes are heavily influenced by intratumour heterogeneity and may therefore not in fact represent different subtypes of disease. Clearly these issues are beyond the scope of the discussion in this manuscript but it is eye opening and will warrant following up.

Minor points:

1. Figure 2: It might be useful if the cell type category of each marker (e.g. CD44: stromal cell marker) is explained. **Agreed, we have expanded the figure legend.**
2. Figure 3A: The “LOH” and “Somatic” should be defined. **Agreed, we have expanded the Methods section. These are classes of variant as defined by the VarScan algorithm which is referred to and cited in the Methods.**
3. Figure 3B: The column labels are overlapped. **Good spot, we have redrawn this figure.**
3. Figure 4B: Each score should be defined. **Agreed, we have expanded the Methods.**
4. Figure 5C: “Fold enrichment” is not defined. **Agreed, we have updated the Methods to describe that the GOTERM analysis was done. Fold enrichment is a DAVID parameter calculated by $(m/n)/(M/N)$ where N is all genes, M is all genes in a given pathway, n is the gene list of interest, and m is the genes in the gene list that belong to the given pathway.**
5. Figure 5D: What does the edge mean? How is the network constructed? **Another good point. We have updated the Methods to describe how the network analysis was done.**

Reviewer #4 (Expertise: CIN, cell cycle, cancer, Remarks to the Author):

Reviewer 4 is also very complimentary, stating that the manuscript consists of “*technically very challenging experiments*”, “*the data seems very solid*”, it is “*presented in a pleasant and clear manner*”, involving “*latest technologies... not applied extensively*” in this context. The critique is fair raising a number of points which we have now addressed, either with new data, changes to the presentation on in the comments below. We thank the reviewer for highlighting areas of the work that required strengthening.

In this manuscript, the authors describe a pipeline for tumor cell isolation, ex vivo culturing and characterization for ovarian cancers. This ‘living biobank’ was further exploited to test the behavior of ex vivo cultured tumor cells in mitosis, cell fate and to assess their drug sensitivities. This paper consists out of technically very challenging experiments and the data seems very solid and are presented in a pleasant and clear manner. It involves some of the latest technologies, including single cell sequencing for both DNA and transcriptome, techniques that have not been applied extensively on these type of tumors/cell lines. Setting up living biobanks will be an important aspect of future patient treatment plans and therefore the development and investigation of the feasibility and reliability of such databanks is of great importance. Besides testing drug sensitivities to allow for the predication of therapy responses, these databanks could also provide us with novel biological findings on tumor behavior. Especially, data on the exact extents of instability in tumors and exact numbers on tumor cell heterogeneity are sparse.

Firstly, I highly appreciate the amount of work and the solidity of the data itself. However, I feel that some of the conclusions that the authors draw from this system are a bit premature. Most importantly, I believe the authors missed the opportunity to show that their ex vivo cultures are indeed reflecting their respective tumors in terms of karyotypes, genomes and mitotic behavior. Which is an important feature for future exploitability and for the validity of their main conclusions. IF the authors can address this point, I would support publication in Nature Communication. I will present my main major and minor issues below.

1. In the current manuscript there is no evidence that this pipeline reliably reflects the characteristics and behavior of the tumor itself. If (for at least a small subset) of these patients

there is material available from the solid tumor for example from solid biopsies or after surgical resection, this would allow the authors to test the parallels between their system and the original tumor. The fact that the two lines derived from the same biopsy (38a,b) are extremely dissimilar could potentially reflect tumor heterogeneity but could also reflect selection under culture conditions. It is unclear why 38a is not tested in many assays. It would be interesting to see if they have common driver mutations, CIN levels and similar drug sensitivities. If behavior is very different, this could again reflect tumor heterogeneity but could also highlight the potential risks underlying the establishment of ex vivo culturing.

As outlined above, we now include data derived from analysis of the primary tumour blocks (Figure S1A and B). In brief, we obtained archival material for 8 of the 12 patients, and performed IHC, staining for Pax8, p53, WT1 and CK7. As part of a tumour gene panel, TP53 was also analysed by targeted amplicon sequencing using a clinical diagnostic service. The congruence between the data derived from the archival blocks and the OCMs is excellent. Also, the IHC analysis of 59 revealed extensive nuclear atypia, consistent with rampant CIN *in vivo*. We have added the analysis of archival blocks to Figure S1A, and added IHC images (Figure S1B). The timelines in Figure 1A have been updated to show when the primary tumours were sampled. Thus, the bottom line is that the OCMs do indeed appear to reflect the primary tumours. Beyond that, we would argue that our work compares favourably with several other recent living biobank papers, e.g. three from the Clevers' lab published in *Cell*, *Cell* and *Nature Medicine* (Kopper et al., 2019; Sachs et al., 2018; van de Wetering et al., 2015), and several observations suggest that our pipeline does indeed generate relevant models of the disease. Firstly, we are building on the work of Ince et al who showed very elegantly that the OCMI conditions yields models that reflect and maintain the genomic and transcriptomic landscape of the primary tumour, as well as the morphological characteristics when grown as PDX (Ince et al 2015). Secondly, we present compelling evidence that the models generated here bare all the expected hallmarks of ovarian cancers. And finally, the killer piece of evidence that the OCMs do indeed reflect their respective tumours comes from the cisplatin sensitivities. OCMs with the lowest IC₅₀ values were derived from patients who achieved a partial response and a reduction in CA125. By contrast, OCMs with a high IC₅₀ originated from patients who demonstrated progressive disease. Regarding OCM.38a, this was one of our first cultures and unfortunately it did not recover from frozen, hence it was not analysed in the latter assays. When we went back to the stocks of unseparated cells, we derived OCM.38b. The exact relationship between 38a and 38b is unclear but incredibly interesting in that it could reflect tumour heterogeneity or it could reflect evolution in culture.

2. Related to the first point, there is no evidence provided that the mitotic behavior observed in this ex vivo culture system reliably reflects the mitotic behaviors in vivo. The 2D ex vivo culturing of these cells might bring extra stresses for these cells that promote segregation errors as these cells are taken out of their 3D context (Knouse et al. 2018) and might even explain the differences observed between these ex vivo cultures and the well established cancer cell lines that have adapted to 2D culturing conditions. And thus, their conclusion that established cell lines underestimate the mitotic errors that occur in tumors is premature and can only be drawn if data is presented that shows that the ex vivo cultures do actually reflect the mitotic behavior of tumors more reliably (which I understand is a challenging request, so at least their conclusions need to be adjusted and discussion on this matter needs to be included).

The reviewer raises a number of interesting points. Regarding their first point, to be fair, we never claimed that the mitotic behaviours observed *ex vivo* reflect the mitotic behaviours *in vivo*. Indeed, analysing the dynamics of chromosome segregation in a cancer patient would indeed be a “*challenging request*”. The point we do make is that analysing established cell lines does not reflect what happens in tumour cells fresh out of a patient – the data to support this is compelling. Having said that, HGSOC is characterized by rampant CIN, this is crystal clear from the various cancer genomics projects, and there must be a mechanistic basis that is inherent to the tumour cells, so should we be surprised that this CIN continues to manifest *ex vivo* in the form of highly abnormal mitoses?

The second issue is more proportionate; is the mitotic chaos we observe a result of 2D

culture stress? While this may seem a reasonable question at first glance, upon reflection it is highly unlikely. Indeed, we controlled for this issue specifically by analysing mitosis in the stromal cells. Like their matched tumour cells, these cells were isolated from the same patient at the same time, grown in 2D in the exact same media in the exact same tissue culture incubator on the exact same plastic surface, transduced with the exact same GFP-H2B lentivirus and analysed on the exact same microscope. Importantly, over 90% of mitoses in the stromal cells are overtly normal.

The reviewer is correct to raise Knouse *et al.* They reported approx. 6-8% lagging chromosomes in non-transformed mouse cells that had been dissociated. This relatively minor defect is similar to what we see in the stromal cells, both in terms of the numbers (~9%) and the phenotype (lagging chromosomes). By contrast, the mitotic abnormalities we observe in the tumour cells are incredibly pervasive, both qualitatively and quantitatively. Knouse *et al.* also showed that segregation fidelity was improved when the cells assembled into a 3D structure ascinar structure. While organoids developed from non-transformed fallopian tube cells assemble into cysts, ovarian cancer cells do not (Hill *et al.* 2018). So it is extremely unlikely that the catastrophic mitoses we observe are accounted for by the cells being grown in monolayers.

Nevertheless, inspired by the reviewer's comment, we set out to test this directly and we now include new data, analysing mitosis in a 3D setting (Figure 9). Specifically, we show that when OCM.66 is grown in 3D, the frequency of mitotic errors is very similarly to that seen in the 2D setting. But what is interesting is that we now also see an additional class of errors that we did not see in 2D. In particular, in several cases we observed chromosomes being ejected from the spindle at the onset of anaphase.

Regarding cell lines, again the reviewer inspired us to directly compare the OCMs with a panel of 8 established ovarian cancer cell lines. We show that they are fundamentally different (Figure 7D). In brief, we used spindle pole numbers during mitosis as a proxy for CIN. Consistent with the stromal cells being diploid and mitotically stable, they typically have 2 spindle poles. Consistent with the tumour cells being highly aneuploid and mitotically unstable, they have abnormal numbers of spindle poles. By contrast, despite the established cell lines having abnormal genomes, they typically have 2 spindle poles. This is entirely consistent with a process whereby the long periods in cell culture that gives rise to established cell lines selects for relatively fit and fast growing subclones that are mitotically relatively stable. And again, this underpins the significance of our work. By describing a workflow that allows us to study ovarian cancer cells that are relatively fresh out of a patient, we open up new opportunities for discovery science and translational research.

3. The original paper describing the OCI medium for the culturing of ovarian cancer cells *ex vivo* (Ince *et al.* 2014 Nat Comm) already performed some overlapping analysis; they have shown that drug responses in OCI cultured cell lines largely correlate to patient responses, they have performed transcriptomics and LOH analysis and some basic characterization of their cultured cell lines (p53 mutations, protein expression). So, the novelty of the current manuscript largely lies in the chromosomal instability part/mitotic behavior. As there is no evidence provided that the CIN of the *ex vivo* cultures reflects that of the tumor (see previous point), this makes me question the overall impact of this manuscript if there is no extra evidence provided on this part. Something that could also enhance the impact is if the authors would attempt to correlate mitotic behavior, CIN levels etc. to drug response. If correlations can be identified, this is an extra argument for the importance and relevance of such living biobanks.

We are somewhat confused by this point. As the reviewer points out, we are using the exact same methodology as Ince *et al.* published previously in Nat. Commun. In that paper they showed very elegantly that the cultures reflected and maintained the genomic and transcriptomic landscape of the primary tumour. Indeed, it is for this reason that we followed their methodology. As we show, the models we establish all have the hallmarks of HGSOc. It is unnecessary and more than a bit unfair to ask us to re-invent the wheel and repeat the entire Ince paper. Moreover, as discussed above, it's not obvious how one would show that the CIN in the tumour reflects the mitoses in the cultures. How would one perform time-lapse microscopy in the tumour without first establishing some form of *ex*

***vivo* culture. We would of course like to correlate mitotic behaviour, CIN levels etc. to drug response but I hope the reviewer will agree that this goes well beyond the scope of this initial breakthrough showing that (a) we can generate models with extensive proliferative potential that are amenable to high resolution cell biology, and (b) that the levels of mitotic chaos are unprecedented when compared with established cell lines.**

Minor issues

1. The timing between treatment and biopsy is highly variable between patients. For example, the sample of patient 38 was treatment naive and only received cisplatin after the ascites collection. In contrast, patient 33 and 69 received platinum relatively long before ascites collection and patient 64 and 74 in between cisplatin treatments. This means that the treatment response was sometimes based on a previous treatment and sometimes on a treatment following cell line establishment. Can the authors indicate these differences?

The reviewer makes an important point; despite there being a “standard of care”, patients are treated on a case-by-case basis. Thus the variability s/he refers to is a reflection of the disease in real life. We can only collect samples when the patients come in for a drain or surgery, and the drug scheduling is decided by the clinical team. We presented the patient time lines in Figure 1 to illustrate this.

With regard to the specific point, patient 38 never received cisplatin, this patient received two lines of carboplatin and paclitaxel, the second line used to treat relapsed, platinum-sensitive disease in which the progression free interval between platinum therapies is more than 6 months. The objective response rates for rechallenge platinum therapy in platinum-sensitive disease is around 40-50% (Figure 1A). It was not unsurprising therefore that this patient had a radiological and biochemical response to rechallenge platinum. Their ascites was sampled prior to rechallenge platinum, as this was the clinical sign that instigated a restaging CT scan that then showed relapsed disease.

For patient 64, the ascites was sampled during dose-dense weekly paclitaxel therapy, which is considered a standard of care therapy in patients with platinum-resistant disease. As is evident on the timelines, the interval between the last platinum-based therapy and the subsequent dose-dense weekly paclitaxel was <6 months, indicating platinum-resistant disease. Furthermore, the radiological response to prior carboplatin-caelyx treated was progressive disease, again suggesting platinum resistant/refractory disease.

For patient 33, the response to second line carboplatin-gemcitabine was modest, with only stable disease seen radiologically, but a biochemical (CA125) response. It is likely that the addition of bevacizumab as a maintenance therapy controlled the tumour growth, but then subsequent CTs scans showed progression. Given the prior platinum based chemotherapy did not lead to a radiological response (only stable disease) the clinical decision was likely to not pursue further potentially toxic platinum-based therapy. Therefore the only platinum-response available is the prior response (stable disease).

Patient 74 was treated with multiple lines of chemotherapy (>10) indicative of HR-deficient high-grade serous ovarian carcinoma. Indeed, exome sequencing revealed a *BRCA1* variant in keeping with HR repair deficiency. Each one of her platinum-based therapies, dating back to her original diagnosis, had led to a radiological or biochemical response, which is presumably why the patient was subsequently rechallenged with platinum-containing regimens. The clinical history would allude to the fact that this patient developed highly platinum-sensitive disease. Interestingly, for patient 74, the shortest treatment free interval between chemotherapies was following dose-dense weekly paclitaxel, suggesting underlying taxol resistance.

2. Patient 64 is a bit confusing. The first ascites collection (64-1) was during a taxol treatment while the third collection (64-3) was shortly followed by a cisplatin treatment. Yet, for 64.1 a cisplatin response (negative) is noted while for 64.3 a taxol response is reported. Are the colours of the treatments swapped? Or the data in the table?

Patient 64 is very interesting and here we have compelling evidence of intra-tumour phenotypic heterogeneity. As above, the chemotherapy schedule was based on clinical

needs of the patient prior to sample collection and culture establishment. We have double checked the colours and the figure is correct.

3. For patient 61, Figure S1 identified a TP53 mutation by sanger while this was not confirmed by exome sequencing. What could explain this discrepancy?

We thank the reviewer for looking at this data carefully as it has prompted us to double check everything. I am pleased to say that our initial analysis was robust – the issues the reviewer highlights are due to the realities of exome sequencing; it is not infallible. Like any technique there are detection threshold issues and the final output is heavily dependent on the algorithms that interpret the raw data. In this case, Figure 3B originally only showed the variants that the VarScan algorithm deemed to be high confidence somatic mutations. Manually digging into each and every variant is laborious and only warranted if there is a pressing need, i.e. if the result is likely to influence the interpretation. Despite these limitations, the results were on the whole very satisfying. The exome data is largely consistent with the Sanger sequencing in terms of TP53 mutations, and consistent with what we know about HGSOC, namely that beyond TP53 there are few other mutations. The exception to the rule in this case was 87 which has a very elevated mutation rate and partial p53 response. So in other words, the genotyping data, both Sanger and exome, served its intended purpose, namely to provide us with confidence that the OCMs are indeed derived from ovarian cancers,

With regard to the specific question about 61, the hotspot R175H missense mutation identified by Sanger sequencing was unambiguous so our assumption was that the exome sequencing simply didn't pick it up, perhaps due to a coverage issue. We have now dug deeper into the data and visualised the sequence read using IGV. This does identify the R175H missense mutation as a low confidence LOH event. In other words, rather than a coverage issue, it is an algorithm issue. We have modified the figure to show this.

4. The two sequential biopsies from patient 74 show overlapping mutations but also 3 extra mutations in different genes for the first biopsy as compared to the 3rd biopsy that was taken very soon after the initial one. How do the authors explain this?

As above, this all comes down to the depth of coverage and/or how VarScan classifies the variants. In 74-3, there was insufficient coverage of MECOM; NF1 was called as a high confidence LOH event; and TSC2 was called as a high confidence germline variant. We have modified the figure but again, nothing substantial in terms of the interpretation or the thrust of the story has changed.

5. Figure 2A-C, it is unclear to which cell line these images/graphs belong to.

Agreed, we have re-written the legend

6. Figure 5C, do the 'overdispersed' genes involve both up- and downregulated genes? Why are they not depicted separately, would this not be more informative?

Perhaps the Reviewer has misunderstood this. By definition, the overdispersed genes are the ones where the standard deviation is greater than 3, i.e. they are substantially up and down relative to the mean for the population. Whether they are up or down relative to some other population isn't the point of the analysis. For the Reviewer's benefit, we show the heat map in Figure R4 which confirms this.

7. Figure 6B, only 1 stromal cell line is assessed. Which one is this and is the behavior of other stromal cells very consistent? Could the authors analyze a few additional stromal cell lines?

This was not clear from the way we presented the figure but the data we show is in fact combined from the analysis of three different stromal cell lines. We now indicate this in the legend.

8. Figure 6C. Cancer cell lines often display problems in spindle orientation. When cells do not divide perfectly perpendicular to the substratum they grow on, anaphase behavior is very difficult to score as the metaphase will seem very broad. Could these types of anaphases maybe be scored as 'unaligned' anaphases and thus explain the presence of such events despite a functional SAC?

An interesting possibility but based on our 20 year's experience analysing mitosis, the time-lapse sequences are pretty unequivocal – despite the SAC apparently being functional, we observe anaphases with unaligned chromosomes.

9. Figure 8B-C, indicate in the figure that these figures relate to OCM59

Agreed, this is now made more clear in the legend.

10. Page 8, line 196; could the overdispersion of mitotic genes not also reflect a higher fraction of proliferative cells?

A good question and we have now added an additional panel to Figure 6 to show that the cells expressing the mitotic genes more highly are those that are classified as likely to be in G2/M by a separate cell cycle analysis. This is encouraging as it shows that the two bioinformatics analyses are congruent and further supports the notion that the tumour cells are indeed progressing through the cell cycle.

References:

- Ahmed, N., and Stenvers, K. L. (2013). Getting to know ovarian cancer ascites: opportunities for targeted therapy-based translational research. *Front Oncol* 3, 256.
- Cancer Genome Atlas Research, N. (2011). Integrated genomic analyses of ovarian carcinoma. *Nature* 474, 609-615.
- Hanahan, D., and Weinberg, R. A. (2011). Hallmarks of cancer: the next generation. *Cell* 144, 646-674.
- Kipps, E., Tan, D. S., and Kaye, S. B. (2013). Meeting the challenge of ascites in ovarian cancer: new avenues for therapy and research. *Nat Rev Cancer* 13, 273-282.
- Kopper, O., de Witte, C. J., Lohmussaar, K., Valle-Inclan, J. E., Hami, N., Kester, L., Balgobind, A. V., Korving, J., Proost, N., Begthel, H., *et al.* (2019). An organoid platform for ovarian cancer captures intra- and interpatient heterogeneity. *Nat Med* 25, 838-849.
- Lin, K. K., Harrell, M. I., Oza, A. M., Oaknin, A., Ray-Coquard, I., Tinker, A. V., Helman, E., Radke, M. R., Say, C., Vo, L. T., *et al.* (2019). BRCA Reversion Mutations in Circulating Tumor DNA Predict Primary and Acquired Resistance to the PARP Inhibitor Rucaparib in High-Grade Ovarian Carcinoma. *Cancer Discov* 9, 210-219.
- Patch, A. M., Christie, E. L., Etemadmoghadam, D., Garsed, D. W., George, J., Fereday, S., Nones, K., Cowin, P., Alsop, K., Bailey, P. J., *et al.* (2015). Whole-genome characterization of chemoresistant ovarian cancer. *Nature* 521, 489-494.
- Sachs, N., de Ligt, J., Kopper, O., Gogola, E., Bounova, G., Weeber, F., Balgobind, A. V., Wind, K., Gračanin, A., Begthel, H., *et al.* (2018). A Living Biobank of Breast Cancer Organoids Captures Disease Heterogeneity. *Cell* 172, 373-386 e310.
- Tohill, R. W., Tinker, A. V., George, J., Brown, R., Fox, S. B., Lade, S., Johnson, D. S., Trivett, M. K., Etemadmoghadam, D., Locandro, B., *et al.* (2008). Novel molecular subtypes of serous and endometrioid ovarian cancer linked to clinical outcome. *Clin Cancer Res* 14, 5198-5208.
- van de Wetering, M., Francies, H. E., Francis, J. M., Bounova, G., Iorio, F., Pronk, A., van Houdt, W., van Gorp, J., Taylor-Weiner, A., Kester, L., *et al.* (2015). Prospective derivation of a living organoid biobank of colorectal cancer patients. *Cell* 161, 933-945.
- Way, G. P., Rudd, J., Wang, C., Hamidi, H., Fridley, B. L., Konecny, G. E., Goode, E. L., Greene, C. S., and Doherty, J. A. (2016). Comprehensive Cross-Population Analysis of High-Grade Serous Ovarian Cancer Supports No More Than Three Subtypes. *G3 (Bethesda)* 6, 4097-4103.

Figures for Reviewers only:

Figure R1. Karyotyping, related to Figure 6. scWGS analysis of 5 different stromal cultures. While this identifies a small number of aneuploidies, e.g. chromosome 7 tetrasomy in 38-3Sb, the karyotypes do not exhibit the highly rearranged and unstable genomes typical of the cancer cells. The HGSOC cell line, Kuramochi, is shown for comparison.

Figure R2a. Comparison of stromal and tumour cultures, related to Figure S1. Phase contrast images of separated stromal and tumour fractions in culture flasks and taken using the microscope used for routine cell culture. Note the very different morphologies of the fibroblastic stromal cells versus the epithelial-like tumour cells.

Figure R2b. Analysis of immunomarkers, related to Figure S1. Immunofluorescence images of cells in separated stromal and tumour fractions stained to detect the fibroblast marker Vimentin (purple) and the DNA (green).

Figure R3. Population RNAseq, related to Figure 3. Cluster analysis and heat map of the OCM tumour samples compared to the mesenchymal and proliferative subtypes. Genes from the two-cluster classification described by Way *et al* (2016) were assigned as over-expressed (+1) or under-expressed (-1) in each cluster. These genes were then matched to genes from the OCM tumour samples and the resulting matrix was used to generate the heat map.

Figure R4. scRNAseq: analysis of overdispersed genes, related to Figure 4. (A) For the 80 tumour cells, we calculated a mean and standard deviation for each gene. The histogram shows the distribution of standard deviations. Genes with SDs greater than 3 were considered overdispersed. **(B)** Heat map showing the read counts for the 32 mitosis-related genes in the 80 tumour cells, with the standard deviation on the right.

Figure R5. M-FISH karyotypes, related to Figure 7. Quantitation of chromosome numbers in OCMs 38b, 59, 66-1 and 79, compared to HCT116, a near diploid colon cancer cell line. This data has been removed from the main figures due to space constraints.

Reviewers' comments:

Reviewer #1 (Remarks to the Author):

The authors fully addressed my previous comments. I feel the paper is ready for publication.

Reviewer #2 (Remarks to the Author):

In this manuscript the authors aim to describe the workflow to generate ovarian cancer models (OCMs), characterize the models and demonstrate their potential to study chromosome instability and drug sensitivity.

Overall some of the details remain vague even though clarification was requested by the reviewers. Extensive characterization is performed. However, some analyses are performed only a single or small subset of cultures, so that results are not generalizable; new cultures are introduced with no annotation.

Specific comments

Please add that ascites was a source of the ex vivo cultures to the Summary / Abstract.

Comparison with primary tumour. IHC was performed on blocks from 8 of the 12 patients from whom OCMs were established. PAX8 was concordant in 7/8 (ie. positive or negative) and p53 mutation in 7/7 tested and IHC in 4/8. This adds some confidence that the OCMs reflect the primary tumour, but it is not a validation. Please change wording.

Note – Table S2 and Figure S1 have different results for OCM33 p53 mutation.

The OCM establishment efficiency is still not stated. The Results now state that “we collected 312 specimens ...” and later that “... we have generated 73 ex vivo cultures”. Is the percentage success 73/312, ie 23%. Please include.

Please include the definition that was used to define an OCM and an OCM as tumour or stromal. Figure S1A includes OCM61 which was not able to be recovered from frozen stock and OCM69 which is stromal. These need to be removed or annotated throughout to distinguish them. As stated in the original reviews not all OCMs have the tumour markers expected and some have strong expression

of stromal markers. Figure 2D shows the flow analysis for 2 epithelial / tumor markers and 2 stromal markers. However, only 2 of the OCMs have the pattern shown (Figure S1A) and it is therefore not typical or representative. There is a description of some of the results under “Characterisation of ex vivo models”. However, a definition of an OCM, tumour and stroma is required. Additionally, given the acknowledged heterogeneity, the clarity of the manuscript and understanding of the downstream analysis would be improved by including a summary of the OCM characteristics in the main manuscript (rather than SI) and annotation to help the reader associate and understand the significance of the downstream results.

Figure 3A. The associated Method p26 states “... somatic when the variant is present in both cell types or loss of heterozygosity when the variant is homozygous in the tumour cell sample only.” This is incorrect. Somatic variants occur only in the tumour cells whereas germline variants occur in all cells. Loss of heterozygosity is loss of the wild type allele.

Figure 3B. The mutation types for p53 do not all match with what is presented in Figure S11.

Page 8, lines 189 – 190 – should the figures referred to be Figures 4 E,F G?

Figure 7: Three OCM additional models are included here. However, they are not annotated, although a reference is provided, and the justification for adding these here and only here is not clear. Additional ovarian cancer cell lines are also included. They are listed Methods, but the sources are not detailed.

A new experiment is included – Figure 9 – but results are only shown for 1 OCM so it is not possible to generalize.

Figure 6 and Figure S3. The source of the HCT cells needs to be included. The genome wide chromosome copy number profiles are shown in Figure 6 with the overflow in Figure S3. However, the placement of results makes it hard to assess; eg, 38a is in Fig 6 and 38b is in Fig S3, 64-3+/- are in Fig 6 and 64-5 is in Fig S3.

Figure 8 and Figure S4. The results are again divided and again results for OCMs derived from the same patient are separated - 74-1 is in Figure 8 but 74-1 is in Figure S4.

Figure 10 and Figure S5. Again, the results are divided; in this case 66-1 is represented in each Figure, but the graphs are different for the 2 sets of results labelled 66-1. One of these may be 66-5.

Reviewer #3 (Remarks to the Author):

The revised manuscript addressed most of my concerns except the following points:

1. Figure 5: A similar analysis showing the high gene expression noise of mitotic genes and the relationship between mitotic gene expression and cell cycle stages (Figure 4E and 4G) should be performed in the new 10X data.

2. Line 189-190 at page 8: "Figure 6E, F" and "Figure 6G" should be "Figure 4E, F" and "Figure 4G", respectively.

Reviewer #4 (Remarks to the Author):

The authors have adjusted the manuscript and have addressed most of the comments.

I have some small textual suggestions but once addressed, I support publication in Nature Communications.

1. In their response, the authors pointed out that they never claim that the mitotic behavior observed ex vivo reflects that of in vivo. Rather, they conclude that the mitotic behavior observed in cell lines is different than that observed in their ex vivo cultures, a claim that is indeed supported by their data. However, I find the sentence in their summary "...indicating that analysis of established cell lines underestimates mitotic dysfunction in advanced human cancers" rather misleading. This sentence does suggest that their ex vivo cultures reflects the behavior in tumours. I therefore suggest to change this sentence as such that it reflects their data: they should/can point out the observed discrepancy between ex vivo established cultures and cell lines and that this MIGHT reflect an underestimation of mitotic dysfunction in advanced human cancers.

2. The authors now also perform a new trick where they prevent proliferation of the stromal cells in a very early passage by the addition of Nutlin-3. This is a neat trick and allows very early analysis of the ex vivo cultures. This however now also provides evidence that in certain cases, some selection takes place upon passaging of the cells. Moreover, their 2D versus 3D culturing was done only for 1 cell line and although only small differences are observed in anaphase behavior, mitotic timing is quite a lot shorter in 3D (which is mentioned), at least for this cell line. This indicates that ex vivo culturing conditions might impact the behavior of the cells to some extent and might induce some selection bias over time. I agree that this system is definitely superior to cell lines and will form a valuable source in the future, however the authors should also mention the potential caveats/limitations in their discussion and maybe stress that it is optimal to analyze the cultures as early as possible, something that they now even have optimized with the Nutlin-3 trick.

3. I appreciate the extra work the authors did to add the data on the primary tumors. However, the authors claim that it is basically impossible to address mitotic dysfunction in the primary tumors. I do not agree with this statement: FISH analysis of one or a small selection of chromosomes or analysis

of anaphase figures from the H&E stainings are technically not that challenging on patient material and would have provided some evidence for similar, high levels of heterogeneity within the primary tumor. This is something that the authors could consider to implement in future studies.

NCOMMS-18-35439A-Z – Response to second round of comments

We thank the four reviewers for taking the time to evaluate our revised manuscript. We were of course very happy to see that all four were largely satisfied with how we addressed their comments. Below we address the few remaining minor issues and detail the changes we have made to the text to address these issues.

Reviewer #1 (Remarks to the Author):

The authors fully addressed my previous comments. I feel the paper is ready for publication.

Once again, we thank Reviewer #1 for their positive response.

Reviewer #2 (Remarks to the Author):

In this manuscript the authors aim to describe the workflow to generate ovarian cancer models (OCMs), characterize the models and demonstrate their potential to study chromosome instability and drug sensitivity. Overall some of the details remain vague even though clarification was requested by the reviewers. Extensive characterization is performed. However, some analyses are performed only a single or small subset of cultures, so that results are not generalizable; new cultures are introduced with no annotation.

We thank Reviewer #2 for their detailed reading of our revised manuscript and are particularly grateful for picking up a number of minor errors. In addition, we have now provided the few remaining details requested. Regarding the extent of the analyses, as summarised in Table R1, the vast majority of the OCMs were subjected to the vast majority of approaches, so in that sense the results are “generalizable”. Beyond that, as we outlined in our first response, HGSOE is an incredibly heterogeneous disease with only two recurrent features, namely near-ubiquitous *TP53* mutation and rampant chromosome instability. Our results, which are supported by incredibly compelling datasets, are entirely consistent with this. Regarding new OCMs, they are annotated in Pillay et al (Cancer Cell, 2019); re-presenting the same data would be inappropriate.

Specific comments:

Please add that ascites was a source of the ex vivo cultures to the Summary / Abstract.

We have now modified the Abstract to mention ascites.

Comparison with primary tumour. IHC was performed on blocks from 8 of the 12 patients from whom OCMs were established. PAX8 was concordant in 7/8 (ie. positive or negative) and p53 mutation in 7/7 tested and IHC in 4/8. This adds some confidence that the OCMs reflect the primary tumour, but it is not a validation. Please change wording.

We have changed the wording and no longer use the word validate.

Note – Table S2 and Figure S1 have different results for OCM33 p53 mutation.

Thank you for pointing out this error, we have corrected Table S1.

The OCM establishment efficiency is still not stated. The Results now state that “we collected 312 specimens ...” and later that “... we have generated 73 ex vivo cultures”. Is the percentage success 73/312, ie 23%. Please include.

We have now included new text in the Discussion to explicitly discuss the percentage success rate.

Please include the definition that was used to define an OCM and an OCM as tumour or stromal.

We have now included definitions for tumour and stromal. Table R2 confirms that these definitions are appropriate.

Figure S1A includes OCM61 which was not able to be recovered from frozen stock and OCM69 which is stromal. These need to be removed or annotated throughout to distinguish them.

We now state in the legend to Figure S1 that OCM61 was not recoverable from frozen stocks. We also now make it clear that OCM69 was overgrown with stromal cells (p6, p16 legend S1) and we annotate it with an asterisk in Figure 3. In their first set of comments, Reviewer #2 asked that we describe the difficulties encountered; therefore we have retained these cultures to illustrate our “real life” experience generating OCMs.

As stated in the original reviews not all OCMs have the tumour markers expected and some have strong expression of stromal markers. Figure 2D shows the flow analysis for 2 epithelial / tumor markers and 2 stromal markers. However, only 2 of the OCMs have the pattern shown (Figure S1A) and it is therefore not typical or representative. There is a description of some of the results under “Characterisation of ex vivo models”. However, a definition of an OCM, tumour and stroma is required. Additionally, given the acknowledged heterogeneity, the clarity of the manuscript and understanding of the downstream analysis would be improved by including a summary of the OCM characteristics in the main manuscript (rather than SI) and annotation to help the reader associate and understand the significance of the downstream results.

As above, we have now included definitions for tumour and stromal. Unfortunately space does not permit moving Figure S1 to the main figures. However, as suggested, to help the reader, we have better annotated Figure 2.

Figure 3A. The associated Method p26 states “... somatic when the variant is present in both cell types or loss of heterozygosity when the variant is homozygous in the tumour cell sample only.” This is incorrect. Somatic variants occur only in the tumour cells whereas germline variants occur in all cells. Loss of heterozygosity is loss of the wild type allele.

We have expanded this text to make it more clear how the VarScan algorithm calls the variants.

Figure 3B. The mutation types for p53 do not all match with what is presented in Figure SI1.

Thank you for pointing this out, 87 has been corrected.

Page 8, lines 189 – 190 – should the figures referred to be Figures 4 E,F G?

Thank you for pointing this out, the figure call outs have been corrected.

Figure 7: Three OCM additional models are included here. However, they are not annotated, although a reference is provided, and the justification for adding these here and only here is not clear. Additional ovarian cancer cell lines are also included. They are listed Methods, but the sources are not detailed.

We have expanded the text to better justify the inclusion of these additional lines and provide additional details. The data is in (Pillay et al) so we have not presented the same data twice as this would be inappropriate. The sources for the ovarian cancer cell lines was already included – see methods.

A new experiment is included – Figure 9 – but results are only shown for 1 OCM so it is not possible to generalize.

The purpose of this experiment was to address a point raised by Reviewer #4; s/he is satisfied with how we addressed their query. Note that the issue was to test the hypothesis that the segregation errors are due to the 2D culture. The fact that a OCM.66 grown in 3D still missegregates negates that hypothesis. We analysed all the OCMs in 2D so our conclusion that they exhibit mitotic chaos is generalizable.

Figure 6 and Figure S3. The source of the HCT cells needs to be included.

We have now included the source of the HCT116 cells.

The genome wide chromosome copy number profiles are shown in Figure 6 with the overflow in Figure S3. However, the placement of results makes it hard to assess; eg, 38a is in Fig 6 and 38b is in Fig S3, 64-3+/- are in Fig 6 and 64-5 is in Fig S3. Figure 8 and Figure S4. The results are again divided and again results for OCMs derived from the same patient are separated - 74-1 is in

Figure 8 but 74-1 is in Figure S4. Figure 10 and Figure S5. Again, the results are divided; in this case 66-1 is represented in each Figure, but the graphs are different for the 2 sets of results labelled 66-1. One of these may be 66-5.

We do of course appreciate that there is an incredible amount of data in this manuscript, and in particular the data sets in Figures 6, 8 and 10 are large and complex. Therefore, to make the data as accessible as possible to readers, we invested a significant amount of effort to present these data sets in a way that allows the reader to see all the data in a visually intuitive manner. But doing so requires keeping the panels reasonably large; presenting all of it in the main figures would require considerable down scaling, thus compromising the presentation. In the main figures therefore we have selected subsets that illustrate the data set as a whole and used the supplementary figures to present the rest so that the reader can see all of the data. This seems like a perfectly rational way to present the data; the co-authors, our local peer reviewers and the three other anonymous reviewers did not object to what is a fairly standard approach. We thank the Reviewer for spotting the labelling error in Figure S5; the OCM is indeed 66-5 so we have corrected this.

Reviewer #3 (Remarks to the Author):

Once again, we thank Reviewer #3 for their constructive comments and helping us improve the manuscript.

The revised manuscript addressed most of my concerns except the following points:

1. Figure 5: A similar analysis showing the high gene expression noise of mitotic genes and the relationship between mitotic gene expression and cell cycle stages (Figure 4E and 4G) should be performed in the new 10X data.

This is an excellent suggestion. And indeed, it was the first analysis we performed on the new scRNAseq data (Figure R1). A brief analysis suggested that there was again high gene expression noise for mitotic genes. However, there are two issues. The first is deconvolving noise from cell cycle stage. In the revised manuscript we showed that cells with high levels of mitotic genes had high G2/M scores (Figure 4G). In retrospect this is somewhat of a self-fulfilling prophecy and that's why we toned down the interpretation, removing the statement suggesting that the overdispersion was a sign of mitotic stress. We had hoped that comparing to the stromal cells would control for this but it raises the second issue, namely heteroscedasticity. If any given set of genes are more highly expressed in the tumour cells, then the standard deviation will be higher so they will appear more overdispersed. Sorting all this out will require a substantial amount of bioinformatics analysis that we hope to do in the future. Indeed, there is a huge amount of information in the scRNAseq dataset that we haven't yet tapped into. But this will require employing a bioinformatician to be embedded in our team. In the meantime, as mentioned we had already modified the interpretation, and saving this analysis for later does not impact the main thrust of this current story.

2. Line 189-190 at page 8: "Figure 6E, F" and "Figure 6G" should be "Figure 4E, F" and "Figure 4G", respectively.

We thanks the reviewer for spotting this error, it has been corrected.

Reviewer #4 (Remarks to the Author):

The authors have adjusted the manuscript and have addressed most of the comments.

I have some small textual suggestions but once addressed, I support publication in Nature Communications.

Again, we would like to thank Reviewer #4 for their constructive comments and helping us improve the manuscript.

1. In their response, the authors pointed out that they never claim that the mitotic behavior observed ex vivo reflects that of in vivo. Rather, they conclude that the mitotic behavior observed in cell lines is different than that observed in their ex vivo cultures, a claim that is indeed supported by their data. However, I find the sentence in their summary "...indicating that analysis of established cell lines underestimates mitotic dysfunction in advanced human cancers" rather misleading. This sentence does suggest that their ex vivo cultures reflects the behavior in tumours. I therefore suggest to change this sentence as such that it reflects their data: they should/can point out the observed discrepancy between ex vivo established cultures and cell lines and that this MIGHT reflect an underestimation of mitotic dysfunction in advanced human cancers.

We have modified the Abstract accordingly.

2. The authors now also perform a new trick where they prevent proliferation of the stromal cells in a very early passage by the addition of Nutlin-3. This is a neat trick and allows very early analysis of the ex vivo cultures. This however now also provides evidence that in certain cases, some selection takes place upon passaging of the cells. Moreover, their 2D versus 3D culturing was done only for 1 cell line and although only small differences are observed in anaphase behavior, mitotic timing is quite a lot shorter in 3D (which is mentioned), at least for this cell line. This indicates that ex vivo culturing conditions might impact the behavior of the cells to some extent and might induce some selection bias over time. I agree that this system is definitely superior to cell lines and will form a valuable source in the future, however the authors should also mention the potential caveats/limitations in their discussion and maybe stress that it is optimal to analyze the cultures as early as possible, something that they now even have optimized with the Nutlin-3 trick.

This is a good suggestion; we have now highlighted the selection pressure and the advantages of the Nutlin-3 selection in the Discussion.

3. I appreciate the extra work the authors did to add the data on the primary tumors. However, the authors claim that it is basically impossible to address mitotic dysfunction in the primary tumors. I do not agree with this statement: FISH analysis of one or a small selection of chromosomes or analysis of anaphase figures from the H&E stainings are technically not that challenging on patient material and would have provided some evidence for similar, high levels of heterogeneity within the primary tumor. This is something that the authors could consider to implement in future studies.

Again, this is a good suggestion and moving forward we would like to perform more experiments on the primary tumour blocks. In particular interphase FISH could yield interesting quantitative data. With regards to mitotic figures, while we do see them (Figure S1B), they are very rare.

Table R1. Summary of analyses applied to the OCM tumour cells

Analyses to validate OCMs; i.e. to determine that the separation process has yielded distinct tumour and stromal fractions							Analyses to characterise OCMs; i.e. experiments to interrogate the genomes, transcriptomes and phenotypes of the tumour models							Bespoke analyses to address specific questions		
OCM	IF	WB	p53 function	TP53 genotyping	IHC	FACS	Exome	RNAseq	scWGS	Spindle poles	Time-lapse	Drug sensitivity	Cell fate	scRNAseq	M-FISH	3D mitosis
33	✓	✓	✓	✓	4	✓	✓	✓	✓	✓	✓	✓	✓	5	5	6
38	✓	✓	✓	✓	✓	✓	✓	✓	✓	✓	✓	✓	✓	✓	✓	6
46	✓	✓	✓	✓	✓	✓	✓	✓	✓	✓	✓	✓	✓	5	5	6
59	✓	✓	✓	✓	✓	✓	✓	✓	✓	✓	✓	✓	✓	✓	✓	6
61	✓	2	✓	✓	✓	✓	✓	✓	2	2	2	2	5	5	6	
64-1	✓	✓	✓	✓	✓	✓	✓	✓	✓	✓	✓	✓	✓	5	5	6
64-3	✓	3	✓	✓	✓	✓	✓	✓	3	3	3	3	5	5	6	
64-3+	✓	✓	✓	✓	✓	✓	✓	✓	✓	✓	✓	✓	✓	5	5	6
64-3-	✓	✓	✓	✓	✓	✓	✓	✓	✓	✓	✓	✓	✓	5	5	6
66-1	✓	✓	✓	✓	✓	✓	✓	✓	✓	✓	✓	✓	✓	5	✓	✓
66-5	✓	✓	✓	✓	✓	✓	✓	✓	✓	✓	✓	✓	✓	5	5	6
69	1	1	1	✓	1	✓	✓	✓	1	1	1	1	5	5	6	
72	✓	✓	✓	✓	✓	✓	✓	✓	✓	✓	✓	✓	✓	5	5	6
74-1	✓	✓	✓	✓	✓	✓	✓	✓	✓	✓	✓	✓	✓	✓	5	6
74-3	✓	✓	✓	✓	✓	✓	✓	✓	✓	✓	✓	✓	✓	5	5	6
79	✓	✓	✓	✓	4	✓	✓	✓	✓	✓	✓	✓	✓	✓	✓	6
87	✓	✓	✓	✓	4	✓	✓	✓	✓	✓	✓	✓	✓	5	5	6

Notes:

1. Once we realised that OCM.69 was overgrown with stromal cells we eliminated it from many of the analyses. However, because DNA and RNA had already been prepared for NGS analysis, we left 69 in as an internal negative control.
2. Unfortunately, OCM.61 – our first solid biopsy derived culture – was not recoverable from frozen stocks so therefore not included in some analyses. We do now have new cultures derived from solid biopsies that are recoverable.
3. Because we separated OCM.63-3 into two subclones, we did not perform several experiments on the mixed culture.
4. Archival blocks were not available from all the primary tumours.
5. scRNAseq and M-FISH were used as orthogonal approaches to support other approaches.
6. 3D analysis of OCM.66-1 was done to test a specific hypothesis.

Table R2. Distinguishing tumour and stromal fractions

OCM	Tumour			Stromal		
	Epithelial morphology	Positive for either Pax8, EpCAM or CA125?	Mutation in TP53 RT-PCR product?	Fibroblastic morphology?	Strong Vimentin staining?	Wildtype TP53 RT-PCR product?
33	✓	✓	✓	✓	✓	✓
38	✓	✓	✓	✓	✓	✓
46	✓	✓	✓	✓	✓	✓
59	✓	✓	✓	✓	✓	✓
61	✓	✓	✓	✓	nd	✓
64-1	✓	✓	✓	✓	✓	✓
64-3	✓	✓	✓	✓	✓	✓
64-3+	✓	✓	✓	na	na	na
64-3-	✓	✓	✓	na	na	na
66-1	✓	✓	✓	✓	✓	✓
66-5	✓	✓	✓	✓	✓	✓
69	x	nd	x	✓	nd	✓
72	✓	✓	✓	✓	✓	✓
74-1	✓	✓	✓	✓	✓	✓
74-3	✓	✓	✓	✓	✓	✓
79	✓	✓	✓	✓	✓	✓
87	✓	✓	x	✓	✓	✓

Notes:

1. When we failed to detect a *TP53* mutation for OCM.69, we looked closer at the culture and the fibroblastic morphology strongly suggested stromal overgrowth so we didn't analyse it for tumour markers (nd). The exome and RNAseq confirmed this assessment.
2. OCMs 74-1 and 74-3 have less striking epithelial morphologies, but they are weakly positive for CA125 and have *TP53* mutations so they were taken forward as tumour cultures. The subsequent NGS-based experiments confirmed this assessment.
3. OCM.87 is a rare *TP53* wildtype but is clearly positive for EpCAM, CA125 and PAX8 so was therefore continued as a tumour culture. The subsequent NGS-based experiments confirmed this assessment.
4. The stromal culture for 61 was not analysed by IF.
5. There are no corresponding stromal fractions for OCMs 64-3+ and 64-3-, just the stromal culture derived from the parental OCM.64-3 culture. Note that OCM.64-3 was mixed for the tumour markers, including EpCAM, allowing us to separate out the two subclones.

Figure R1. (A) Venn diagram showing the number of overdyspersed genes in tumour and stromal cells. (B) Histogram showing the number of overdyspersed genes shared between samples 1, 2, 3 and 4. (C) Network analysis of the overdyspersed "mitotic" genes common to the four tumour samples. (D) Violin plots showing the average read count for the overdyspersed tumour genes showing that they are more highly expressed compared to the stromal cells, illustrating the heteroscedasticity issue.

REVIEWERS' COMMENTS:

Reviewer #2 (Remarks to the Author):

My comments have been addressed. I have no further comments.

Reviewer #3 (Remarks to the Author):

The revised manuscript did not address my previous concern. The authors should show that the main finding of the first single-cell data (the expression noise in tumor cells largely reflects more heterogeneous cell cycle stages) is validated in the new 10X data. This can be easily done by doing the same analysis (PAGODA overdispersion analysis) of Figure 4E for each tumor-stromal pair. The PAGODA was designed to account for the heteroscedasticity issue raised by the authors. The heterogeneity of cell cycle stages in tumor cells compared to stromal cells can be also visualized by plotting the distribution of predicted cell cycle stages of each cell between tumor and stromal cells for each pair.

Reviewer #4 (Remarks to the Author):

The authors have fully addressed all my comments. I now fully support publication in Nature Communications

NCOMMS-18-35439A-Z – Response to second round of comments

We thank the four reviewers for taking the time to evaluate our revised manuscript. We were of course very happy to see that two of the three of the Reviewers whom previously commented now support publication. Below we address the remaining concern of Reviewer#3.

Reviewer #2 (Remarks to the Author):

My comments have been addressed. I have no further comments.

We thank Reviewer #2 for their final review.

Reviewer #3 (Remarks to the Author):

The revised manuscript did not address my previous concern. The authors should show that the main finding of the first single-cell data (the expression noise in tumor cells largely reflects more heterogeneous cell cycle stages) is validated in the new 10X data. This can be easily done by doing the same analysis (PAGODA overdispersion analysis) of Figure 4E for each tumor-stromal pair. The PAGODA was designed to account for the heteroscedasticity issue raised by the authors. The heterogeneity of cell cycle stages in tumor cells compared to stromal cells can be also visualized by plotting the distribution of predicted cell cycle stages of each cell between tumor and stromal cells for each pair.

We agree with the reviewer that the single-cell data requires further analysis. Indeed, as per our previous response, there is a wealth of information in this dataset that we have not yet tapped into. However, doing so will require dedicated bioinformatics support which we are currently setting up. In the future therefore, we hope to be able to address this issue directly and present the findings in further publications. In the meantime, as mentioned we had already modified the interpretation, and saving this analysis for later does not impact the main thrust of this current story. Nevertheless, to acknowledge this point, as per the editor's guidance, we have added a line to the text stating that "*Further analysis will however be required to evaluate the nature of this heterogeneity, including whether or not it reflects differences in cell cycle stage.*"

Reviewer #4 (Remarks to the Author):

The authors have fully addressed all my comments. I now fully support publication in Nature Communications

We thank Reviewer #4 for their final review and positive response.